# Sequential Condition Evolved Interaction Knowledge Graph for Traditional Chinese Medicine Recommendation

## Abstract

Traditional Chinese Medicine (TCM) has a rich history of utilizing natural herbs to treat a diversity of illnesses. In practice, TCM diagnosis and treatment are highly personalized and organically holistic, requiring comprehensive consideration of patients' states and symptoms over time. However, existing TCM recommendation approaches overlook the changes in patients' states and only explore potential patterns between symptoms and prescriptions. In this paper, we propose a novel Sequential Condition Evolved Interaction Knowledge Graph (SCEIKG), a framework that treats the model as a sequential prescription-making problem by considering the dynamics of patients' conditions across multiple diagnoses. In addition, we incorporate an interaction knowledge graph to enhance the accuracy of recommendations by considering the interactions between different herbs and patients' conditions. Experimental results on the real-world dataset demonstrate that our approach outperforms existing TCM recommendation methods, achieving state-of-the-art performance.

## 1 Introduction

Traditional Chinese Medicine (TCM) is an ancient and comprehensive system that has been integral to Chinese society for millennia (Cheung, 2011). TCM differs from Western medicine in light of its unique theoretical foundation, diagnosis methods, and treatment approaches, emphasizing the harmonious functioning of the body's structures (Zhang et al., 2015). Chinese Herbal Medicine, a key component of TCM, has gained global recognition for its positive impact on various illnesses. As a result, TCM recommendation systems, which assist physicians in making informed decisions about prescribing herbs, have emerged as crucial tools. However, TCM practitioners traditionally employ observation, listening, questioning, and pulse-taking methods to understand the overall disease conditions of patients, rather than treating individual symptoms. Furthermore, TCM diagnosis and treatment prescriptions are often based on clinical experience, lacking standardization in sophisticated TCM knowledge. It is, however, essential to note that systems are not intended to replace the expertise of physicians, but rather augment it.

Recently, there have been approaches that perform effectively. However, we found that there are still two shortcomings: (1) **many approaches** (Ruan et al., 2019; Jin et al., 2020; 2021; Yang et al., 2022) **primarily focus on patient symptoms or herbs, neglecting the explicit prediction of how a patient's state may change after taking medication.** *As an example, consider two patients, $x_i$ and $x_j$ (as shown in Fig.1a), both struggling with insomnia, but with different sets of symptoms. Patient $x_i$ presents $sc_1^{(i)} = \{wakefulness, irritability, bitter mouth\}$, while patient $x_j$ has $sc_2^{(j)} = \{dreamy, palpitations, fatigue\}$. Subsequently, both patients took the corresponding herbal prescriptions $hc_1^{(i)}$ and $hc_2^{(j)}$, and the same symptoms set, $sc_3$, appeared at their next diagnosis. Based on the same set of symptoms, the doctor writes the same prescription. However, after the current diagnosis, patient $x_i$ experiences remission, while patient $x_j$ does not. Why is that? The answer may lie in the fact that both patients are in different states —state $o_1$ and state $o_2$ —with the same prescription $hc_3$ not accounting for these variations, potentially undermining the effectiveness of treatment.* While some Western medicine recommendation methods (Yang et al., 2021; Shang et al., 2019; Yang et al., 2022) consider historical data, they do not explicitly predict the patient's post-medication state. (2)

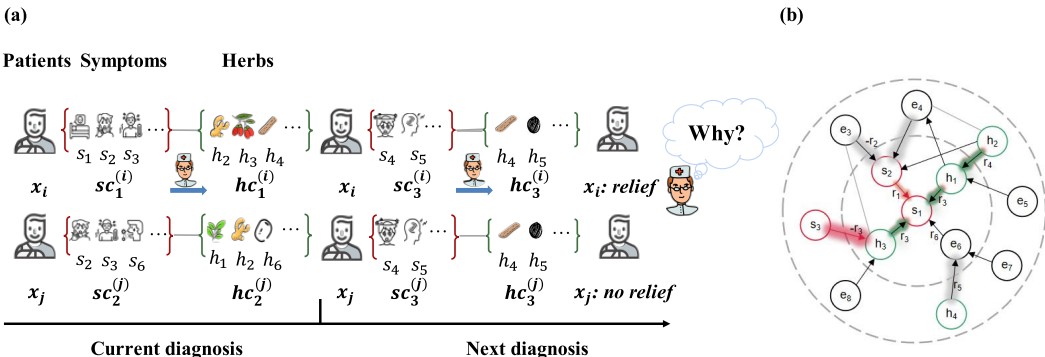

Figure 1: (a) An example of prescribing herbs based on evolution in patient symptoms; (b) An example of IKG containing information about multiple entities.

**Insufficient utilization of domain knowledge.** Most methods (Wang et al., 2019b) typically focus on mining the symptoms and prescriptions within the dataset or incorporate domain knowledge as pre-trained model inputs. However, actual TCM treatment involves four intricate steps laden with profound knowledge. Consequently, relying solely on dataset information falls short of unveiling the complexity of symptom interaction. Additionally, the lack of standardized practices in TCM makes it challenging, and many methods prescribe a fixed set of remedies, which may not be suitable for a patient's condition. *For instance, if a patient describes symptoms such as {headache, runny nose, cough} relying solely on current symptoms provides incomplete information. In reality, these symptoms may also be correlated with other conditions like a sore throat. Hence, depending solely on symptoms from the dataset cannot capture crucial high-level insights. To formulate appropriate herbal prescriptions, richer information is needed, considering the complex associations between symptoms as well as the compatibility between different herbs.* In this way, we can better understand the patient's condition and provide more accurate herbal treatment recommendations.

Motivated by the aforementioned shortcomings, we introduce a novel conceptual framework SCEIKG, which aims to enhance the accuracy of prescribing rational treatments by learning how patients' conditions evolve over multiple sequential diagnoses. Our approach builds upon two key observations: (1) **explicitly leverage on the change in the state of the patient after taking the medication.** We argue that this crucially hinges on the explicit as well as implicit overall condition patient's symptoms described as to why a particular relevant herbal score is coupled to a particular patient. Because each patient has a unique constitution, even when given the same prescription, the resulting changes in their condition can vary widely. Therefore, TCM recommendations must take into account the evolution of the patient's condition. To address this, we introduce a module that predicts how a patient's condition will change after taking medication. This predictive capability enables our model to make reasonable TCM recommendations, even when information about the patient's future state is unavailable. (2) **incorporating domain knowledge for symptom richness and herb compatibility.** We recognize the importance of domain knowledge in ensuring the richness of symptoms and compatibility of herbs. *Based on the example of a patient's consecutive diagnoses, who was suffering from sleepness, bitter mouth, dry throat, etc., we leverage TCM knowledge graph domain knowledge to make extrapolations based on incomplete symptom information. By employing a GNN with IKG as additional auxiliary information, we identify that a specific herb set, including salvia miltiorrhiza and ostrea gigas, can effectively address the symptoms set. This conclusion is drawn from the long-range connections in the graph $s_1 \xrightarrow{r_1} s_2 \xrightarrow{-r_2} e_3 \xrightarrow{r_3} e_4 \xrightarrow{r_4} \{h_1, h_2, h_3, ...\}$. Further, we aggregate high-order similarity relationships and interactions among triplets using graph-based methods, enhancing our understanding of the complex relationship between herbs and symptoms (as depicted in Fig.1b).* Inspired by (Wang et al., 2019a) and (Tu et al., 2021), a hybrid structure, the Interaction Knowledge Graph (IKG), which combines the knowledge graph neighborhood knowledge of TCM and the symptoms-herbs graph to model the intricate relationships between symptoms and herbs. Also, we employ a strategy that involves training both IKG and sequential recommendation models to seamlessly integrate structured and unstructured information. This integrated approach provides a more comprehensive understanding and prediction of real-world scenarios, empowering our recommendation model with dynamic capabilities. Meanwhile, we up-

date the graph structure based on the correlation-based attention mechanism employing the domain knowledge of IKG, which is accomplished by propagating different relation types among entities in the IKG, thus alleviating the issue of herb compatibility to some extent.

We end with a thorough empirical evaluation of our approach to our new collection of real-world data, where we explore the benefits of assessing the condition of the patient after taking medicine. Our results show that learning in a way that accounts for patients' symptoms set and the change of conditions by the sequential diagnoses has significant advantages on TCM recommendation tasks.

## 2 PROBLEM FORMULATION

We denote the set of symptoms by $\boldsymbol{S} = \{s_1, s_2, ..., s_M\}$ and the set of herbs by $\boldsymbol{H} = \{h_1, h_2, ..., h_N\}$, respectively. Note that a symptom $s_i$ is represented by a TCM symptom term, e.g., 抑郁症 *(depression)*; a herb $h_i$ is represented by a TCM herb term, e.g., 茯苓 *(tuckahoe)*. **We define an IKG by** $\mathcal{G} = (\mathcal{E}, \mathcal{R}, \mathcal{T}, \mathcal{A})$**, where** $\mathcal{E}$ **is a set of** *entities* **and** $\mathcal{R}$ **is a set of** *relations*. $\mathcal{T}$ **is a set of triples** $\mathcal{T} = \{(h, r, t) | h \in \mathcal{E}, t \in \mathcal{E}, r \in \mathcal{R}\}$**, where each triples means there is a relation** $r$ **from** *head* **entity** $h$ **to** *tail* **entity** $t$**.** Specifically, $\mathcal{E}$ consists of symptoms $\boldsymbol{S}$, herbs $\boldsymbol{H}$, and other entities such as *pharmacology*, *efficacy*, *diseases*, *examination* and *diagnosis*, which were extracted from TCM datasets (Yao et al., 2018) to help entail relations between symptoms and herbs directly or indirectly (c.f. Appendix A.1). A relation $r \in \mathcal{R}$ indicates the relationship among entities, e.g., *symptoms-related herbs*. The adjacency matrix $\mathcal{A} = [a_{e_i, e_j}]_{V \times V}$ was built based on different types of edge relationships by the co-occurring probabilities using Normalized Pointwise Mutual Information (Bouma, 2009):

$$a_{e_i, e_j} = \begin{cases} 1 - \frac{log(p(e_i)p(e_j))}{log\, p_r(e_i, e_j)}, & \text{if } (e_i, e_j) \text{ co-occur in } \mathcal{T} \\ 0, & \text{otherwise} \end{cases} \tag{1}$$

where $p_r(e_i, e_j)$ is the joint probability of $e_i$ and $e_j$ with relation $r$, and $p(e_i)$ (or $p(e_j)$) is the probability of occurrences of $e_i$ (or $e_j$) in all relations. $V$ is the number of entities in $\mathcal{G}$.

In this paper, we denote the set of patients by $\boldsymbol{X} = \{x_1, x_2, x_3, ..., x_J\}$, and the set of sequences of *diagnoses* by $\Omega = \{\omega_j | \omega_j = \langle \omega_j^{(1)}, \omega_j^{(2)}, \dots, \omega_j^{(T_j)} \rangle, 1 \le j \le J\}$, where $\omega_j^{(t)} = (O_j^{(t)}, \mathbb{S}_j^{(t)}, \mathbb{H}_j^{(t)})$ is the $t$-th diagnosis for $1 \le t \le T_j\}$, and $T_j$ is the number of diagnoses of patient $x_j$. $O_j^{(t)}$ is the description of patient $x_j$ during the $t$-th diagnosis in the form of natural language text, $\mathbb{S}_j^{(t)} \subseteq \boldsymbol{S}$ is a set of symptoms of patient $x_j$ in $t$-th diagnosis, and $\mathbb{H}_j^{(t)} \subseteq \boldsymbol{H}$ is a set of herbs given to patient $x_j$ in $t$-th diagnosis. Our TCM recommendation problem can be formulated by: given a set of sequences of diagnoses $\Omega$ and an initial IKG $\mathcal{G}$, we aim to learn a model $\mathcal{M}_\Theta$ to recommend a set of herbs to a new patient $x_{new}$ in the $k$-th diagnosis based on the patient's historical sequence of diagnoses $\omega_{new} = \langle \omega_{new}^{(1)}, \omega_{new}^{(2)}, \dots, \omega_{new}^{(k-1)} \rangle$, the current description $O_{new}^{(k)}$ and the current symptom $\mathbb{S}_{new}^{(k)}$, i.e.,

$$\mathbb{H}_{new}^{(k)} = \mathcal{M}_\Theta(\omega_{new}, O_{new}^{(k)}, \mathbb{S}_{new}^{(k)}),$$

where $\Theta$ is the parameters of the model to be learned from $\Omega$ and $\mathcal{G}$. $\omega_{new}$ denotes sequential information about the patient and $\omega_{new}^{(k-1)}$ denotes the single diagnosis. Note that $\Theta$ includes both the neural network parameters and the representation parameters of entities and edges in $\mathcal{G}$.

## 3 MODELING APPROACH

In this section, we present SCEIKG, a Sequential Condition Evolved based on an Interaction Knowledge Graph learning model to enhance the accuracy of TCM recommendation. The framework, depicted in Fig.2, consists of three modules. (1) A heterogeneous Graph Neural Network (GNN) utilizes a hierarchical attention message passing layer and knowledge graph embedding layer for entity embeddings. (2) A horizontal condition module learns the patient's current representation from historical records and generates an herbal vector measuring similarity with the patient representation. (3) A transition condition module, which observes the patient's progression after herbal intake, with the evolved status as an auxiliary indicator for subsequent diagnoses. The framework is trained with a joint objective function to ensure accurate TCM recommendations.

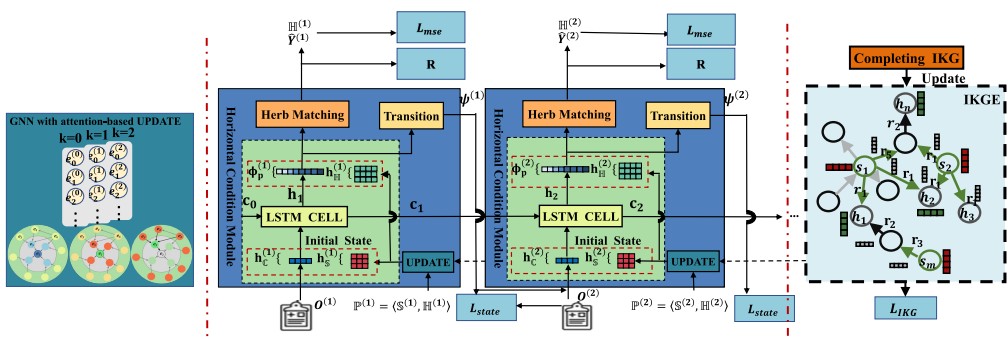

Figure 2: Schematic illustration of the proposed method.

## 3.1 Heterogeneous GNN with attention-based UPDATE

The heterogeneous Graph Neural Network (GNN) updates entity representations through recursive propagation. The $\text{AGGREGATE}^{(k)}(\cdot)$ function integrates the entity $h$ feature from its neighboring $t$ conditioned to relation $r$. This operation is represented as $\alpha_h^{(k)} = \text{AGGREGATE}^{(k)}\left(e_h^{(k-1)}, \sum_{t \in \mathcal{N}(h,r)} w_{(h,r,t)} e_t^{(k-1)}\right)$, where $e_h^{(k)} \in \mathbb{R}^{D_k}$ is the feature embedding of entity $h$ at layer $k$, and $\mathcal{N}(h,r)$ indicates neighbors connected to $h$ with relation $r$. The function propagates integrated information to update entity features at the next layer. To capture higher-order similarities between entities, the interaction knowledge graph (IKG) is utilized. Inspired by KGAT (Wang et al., 2019a), the $\text{UPDATE}(\cdot)$ function updates the weights $\mathbb{W}$ of relations in the IKG, indicating information propagation strength from $t$ to $h$ based on the relationship $r$. The weight value $w_{(h,r,t)} = \text{UPDATE}\left(\{w_{(h,r,t)} \mid (h,r,t) \in \mathcal{G}\}\right) = Softmax\left((W_r e_h + e_r) * W_r e_t\right)$ is calculated using an attention mechanism, considering the correlation between $e_h \in \mathbb{R}^{D_{in}}$ and $e_t \in \mathbb{R}^{D_{in}}$ in the specific-relation $r$ space. The weight value depends on the transformation matrix $W_r$ of relation $r$. The final weight matrix $\mathbb{W} \in \mathbb{R}^{V \times V}$ is obtained from the $\mathcal{A}$ by graph-based Laplcian (Kipf & Welling, 2016) calculation to assess connections between all entities. And the $\text{AGGREGATE}^{(k)}(\cdot)$ and $\text{COMBINE}^{(k)}(\cdot)$ can formulate in the matrix as follows:

$$E^{(k)} = SUM\left(\boldsymbol{NN_1}\left(\left(E^{(k-1)} + \mathbb{W}E^{(k-1)}W_1^{(k)}\right); W_3^{(k)}\right), \boldsymbol{NN_2}\left(\left(E^{(k-1)} \odot \mathbb{W}E^{(k-1)}W_2^{(k)}\right); W_4^{(k)}\right)\right) \tag{2}$$

where $E^{(k)} = [e_1^{(k)}, ..., e_h^{(k)}] \in \mathbb{R}^{V \times D_{k+1}}$ is the stack of entity feature vectors, and $e^{(0)} \in \mathbb{R}^{D_{in}}$ initialization using a uniform distribution. $\boldsymbol{NN_1}(\cdot)$ and $\boldsymbol{NN_2}(\cdot)$ denote forward propagation neural networks with an activation function, $W \in \mathbb{R}^{D_k \times D_{k+1}}$ represents the network weights. The final entity representations $E = Concatenate\left(E^{(0)}, .., E^{(k)}\right) \in \mathbb{R}^{V \times D_{out}}$ are defined by simply concatenating the entity features of all layers, where the $D_{out}$ is the dimension of the embedding space.

Knowledge graph representation enhances link completeness between entities, providing more nuanced information. We use TransR (Lin et al., 2015) combined with RotatE (Sun et al., 2019a) to represent entities, enabling them to play different roles in various triplets to complement links in $\mathcal{G}$:

$$\boldsymbol{f}(h,r,t) = C * ||Sin(W_r e_h + e_r - W_r e_t)||_1 \tag{3}$$

Here, $W_r \in \mathbb{R}^{R \times D_{in}}$ is the transformation matrix of relation $r$ and $C$ is the modulus of constraint, $||\cdot||_1$ denotes the $L_1$-norm. A lower score of $\boldsymbol{f}(h,r,t)$ indicates a higher likelihood that the triplet is true and vice versa. By completing links between entities, we further update the entity representation $E \in \mathbb{R}^{V \times D_{out}}$. The training of TransR combined with RotatE involves considering the relative order between valid triplets and broken ones. It encourages discrimination between them through a pairwise ranking loss, denoted as $L_{IKG}$ (described in detail in Section 4.4).

## 3.2 HORIZONTAL CONDITION MODULE

In TCM, maintaining a harmonious body structure and considering the overall well-being of the patient is paramount. Therefore, it is crucial to gain a comprehensive understanding of the patient's core health state.

**Condition Representation.** To extract the patient's state, we employ Bidirectional Encoder Representations from Transformers (BERT) (Devlin et al., 2018) pre-trained transformer base model. With the exception of fine-tuning transformer models, the condition representation $h_{\mathbb{C}}^{(t)} \in \mathbb{R}^l$ is derived not only from the $l$-dimensional hidden state $h_{bert}^{(t)}$ of the "[CLS]" token in the last layer but also from an average pooling layer $g(\cdot)$ that extracts the overall patient condition vector by assigning weights, thereby focusing on critical and effective information. The process of condition representation can be formulated $h_{\mathbb{C}}^{(t)} = \sum_i^{\Gamma} g(h_{bert}^{(t)}; W_5)_i h_{i,bert}^{(t)}$, where $\Gamma$ is the length of the record sequence $O^{(t)}$, and the average pooling layer $g(\cdot) : \mathbb{R}^{\Gamma \times l} \to \mathbb{R}^l$ combines $h_{bert}^{(t)} \in \mathbb{R}^{\Gamma \times l}$ with assigned attention weights between words $i$ and $j$ to obtain the condition representation $h_{\mathbb{C}}^{(t)} \in \mathbb{R}^l$. A feed-forward neural network, $NN_3(\cdot) : \mathbb{R}^l \to \mathbb{R}^{D_{out}}$, is then applied for dimensionality transformation. To prevent over-fitting, we also apply a high dropout rate to this high-dimensional condition representation.

**Symptoms Representation.** In TCM recommendation, we shift from modeling relationships between single users and items to considering sets of symptoms and herbs. The symptoms set $\mathbb{S}^{(t)}$ is encoded into the multi-hot symptoms $sc^{(t)} \in \{0,1\}^M$, and shared symptoms embeddings table $E_s \in E : \mathbb{R}^{M \times D_{out}}$ explicitly aggregates multi-hop connectivity information related to symptoms, herbs and the similar entity representations in the IKG. We introduce the corresponding symptoms into the embedding space by vector-matrix dot product, represented as $h_{\mathbb{S}}^{(t)} = \sum_{i:sc_i^{(t)}=1}^M sc_i^{(t)} E_{s,i}$, where $h_{\mathbb{S}}^{(t)} \in \mathbb{R}^{D_{out}}$ stores the embedding vector for particular symptoms in the $t$-th diagnosis symptoms for one patient.

**Horizontal Patient Representation.** It is possible that a health snapshot will not be sufficient to make treatment decisions. For example, at the previous $(t-1)$-th diagnosis, patients experienced insomnia, and the prescribed herbs provided relief, leading to the observation of patients in a new state $\Psi^{(t-1)}$. However, at the current $t$-th diagnosis, patients did not mention the insomnia-related symptoms but presented only with a headache. Encoding herbs set $\mathbb{H}^{(t)}$ to multi-hot herbs $hc^{(t)} \in \{0,1\}^N$. We use an LSTM (Hochreiter & Schmidhuber, 1997) model to dynamically model patients' historical states $\Psi^{(t)} = [\Psi^{(t-1)}, \Psi^{(t-2)}, ..., h_C^{(0)}]$ and eventually obtain comprehensive patients representations $\Phi_P^{(t)}$:

$$\Phi_P^{(t)} = LSTM \left( \left( \Pi \left( h_{\mathbb{S}}^{(t)}, \Psi^{(t-1)} \right), C^{(t-1)} \right), ..., \left( \Pi \left( h_{\mathbb{S}}^{(0)}, h_{\mathbb{C}}^{(0)} \right), C^{(0)} \right); W_6 \right) \tag{4}$$

where $C^{(t-1)}$ includes the hidden state $h^{(t-1)} \in \mathbb{R}^{hidden\_dim}$ and cell state $c^{(t-1)} \in \mathbb{R}^{hidden\_dim}$, and the initial state $h^{(0)}$ and $c^{(0)}$ are all-zero vectors, $C$ will be passed down. The State $\Psi^{(t-1)} = T \left( \Phi_P^{(t-1)}, hc^{(t-1)} \right)$ is obtained by transferring the state $\Phi_P^{(t-1)}$ at the $(t-1)$-th diagnosis to the state after taking the herbs $hc^{(t-1)} \in \{0,1\}^N$. A more compact representation $\Pi : \mathbb{R}^{2D_{out}} \to \mathbb{R}^{D_{out}}$ of the patient is created by concatenating the historical condition representation $\Psi^{(t-1)} \in \mathbb{R}^{D_{out}}$ and symptoms representation $h_{\mathbb{S}}^{(t)} \in \mathbb{R}^{D_{out}}$ as a double-long vector, along with a layer of self-attention. Additionally, The operation of transition condition $T(\cdot)$ will be introduced in section 3.3.

**Patient-to-herb Matching.** After obtaining patients' horizontal representation $\Phi_P^{(t)} \in \mathbb{R}^{D_{out}}$, we aim to identify the most relevant herbs from the herbs embedding table $E_h \in E : \mathbb{R}^{N \times D_{out}}$. To achieve this, we perform an inner product to calculate the scores between $E_h$ and $\Phi_P^{(t)}$, followed by the application of the sigmoid $\sigma(\cdot)$ to complete the operation $P(\cdot)$ of herbal recommendation. The operation is denoted as $\hat{Y}^{(t)} = P \left( \Phi_P^{(t)}, E_h \right) = \sigma \left( \Phi_P^{(t)} E_h^T \right)$, where $\hat{Y}^{(t)} \in \mathbb{R}^N$, and each element stores a matching score for one herb. Finally, we obtain the recommended herbal set $\mathbb{H}^{(t)}$ based on

$\hat{Y}^{(t)}$. Finally, we train our model by comparing the loss $L_{mse}$ (described in detail in Section 4.4) of actual and predicted herbs

## 3.3 TRANSITION CONDITION MODULE

In practice, disease treatment is a complex and gradual process, making it challenging for patients to achieve complete recovery with a single treatment. Due to the diverse changes in each patient's status, implicitly representing their post-herb status is essential. The operation of transition condition $\boldsymbol{T}(\cdot)$ is designed to model the shift in patients' conditions after taking herbs $hc^{(t-1)} \in \{0, 1\}^N$. Specifically, we obtain one-dimensional convolution results, $h_{\mathbb{H}}^{(t)} = \boldsymbol{Conv1D}\left(\boldsymbol{P}\left(\Phi_P^{(t)}, E_h\right) E_h; W_7\right)$, to capture global information and eliminate position effect. Inspired by (Liu et al., 2018), herb and patient interactions $h_{\mathbb{I}}^{(t)}$ are considered by multiplying the matrix elements of the herb representation $h_{\mathbb{H}}^{(t)} \in \mathbb{R}^{D_{out}}$ and the horizontal patient representation $\Phi_P^{(t)} \in \mathbb{R}^{D_{out}}$. Finally, one-dimensional convolution results of herb representation $h_{\mathbb{H}}^{(t)}$, interaction representation $h_{\mathbb{I}}^{(t)} \in \mathbb{R}^{D_{out}}$ and patient representation $\Phi_P^{(t)}$ are concatenated into an embedding space to represent transition conditional representations $\Psi^{(t)}$ of patients after taking herbs thus achieving a state transfer. Formally, transition condition module $\boldsymbol{T}(\cdot)$ can be defined by:

$$\Psi^{(t)} = \boldsymbol{T}\left(\Phi_P^{(t)}, \boldsymbol{P}\left(\Phi_P^{(t)}, E_h\right)\right) = \boldsymbol{NN_4}\left(Concatenate\left(\Phi_P^{(t)}, (\Phi_P^{(t)} \odot h_{\mathbb{H}}^{(t)}), h_{\mathbb{H}}^{(t)}\right); W_8\right) \quad (5)$$

where the $\odot$ is the Hadamard product. Note that, the transition condition representation $\Psi^{(t)} \in \mathbb{R}^{D_{out}}$ is represented as the condition representation $h_{\mathbb{C}}^{(t+1)}$ at the next diagnosis. $\boldsymbol{NN_4}(\cdot)$ represents a feed-forward neural network, primarily tasked with performing nonlinear mapping on input data. $W_8$ is the weight parameter of the neural network.

## 3.4 MODEL TRAINING WITH OBJECTIVE FUNCTION

Our approach to robust learning is based on regularized risk minimization, where regularization is to discourage the effects of two mutually exclusive herbs in recommendation. The joint objective is:

$$\underset{\boldsymbol{T}, \boldsymbol{P}, \Theta}{argmin} L_{mse}\left(\boldsymbol{P}^{(t)}, \Theta\right) + L_{state}\left(\boldsymbol{T}^{(t)}; \Theta\right) + \lambda \boldsymbol{R}\left(\boldsymbol{P}^{(t)}, \mathbb{W}, \Theta\right) + L_{IKG}\left(\mathcal{G}, \Theta\right) + \lambda_\Theta ||\Theta||_2^2 \quad (6)$$

where $\boldsymbol{P}$ is the function predicting herbs, $L_{mse} = \sum_{i=1}^{N} (hc_i^{(t)} - \hat{Y}_i^{(t)})^2$ is the average loss w.r.t an empirical MSE loss function. The objective involves evaluating the distance between the recommended herbs set and the ground truth herbs set. And $\boldsymbol{T}^{(t)}$ is a function of transition condition and $L_{state} = Cos(h_{\mathbb{C}}^{(t+1)}, \Psi^{(t)}) = \frac{h_{\mathbb{C}}^{(t+1)} \cdot \Psi^{(t)}}{||h_{\mathbb{C}}^{(t+1)}|| * ||\Psi^{(t)}||}$ measures the cosine similarity between the state $\Psi^{(t)}$ after taking herbs and the next state $\Phi_P^{(t+1)}$. The regularization scheme $\boldsymbol{R} = -\sum_{i=1}^{N} \sum_{j=1}^{N} \mathbb{W}_{ij} \hat{Y}_i^{(t)} \hat{Y}_j^{(t)}$ penalizes $\boldsymbol{P}^{(t)}$ for violating certain pair of herbs, where the $\mathbb{W}_{i,j}$ from the weight $\mathbb{W}$ in $\mathcal{G}$ indicates the strength of compatibility between $i$-th herb and $j$-th herb. If they are mutually exclusive, then $\mathbb{W}_{i,j}^h = 0$.

As we all know, constructing a complete TCM knowledge graph is a difficult task that relies on extensive data support. Therefore, the loss $L_{IKG} = \sum_{(h,r,t,t') \in \mathcal{T}} -ln\sigma\left(\boldsymbol{f}\left(h, r, t'\right) - \boldsymbol{f}\left(h, r, t\right)\right)$ is to complete the TCM knowledge graph, allowing for the inference of useful information that was not initially available. The $\mathcal{T} = \{(h, r, t, t') | (h, r, t) \in \mathcal{G}, (h, r, t, t') | (h, r, t) \notin \mathcal{G}\}$ is the broken triplet constructed by replacing one entity in a valid triplet randomly. Also, $\lambda_\Theta$ controls the $L_2$ regularization strength to prevent over-fitting. Note that the above loss functions are defined for a single diagnosis, and during training, loss backpropagation is conducted at the patient level by averaging the losses across all diagnoses. In section 4, we will demonstrate the effectiveness of these methods in practice, and the detailed algorithmic steps will be presented in Appendix B.

## 4 EXPERIMENTS AND RESULTS

In this section, we present the effectiveness of the model through performance comparisons with various models and additional experimental analyses. Further details on data descriptions, model architectures, training inference, experimental settings, parameter sensitivity, and interpretative experiments regarding herb compatibility and embedding visualization are provided in the Appendix.

Table 1: Performance Comparison on ZzzTCM Dataset.

| Models | Precision | | | Recall | | | F1 | | | |
|---|---|---|---|---|---|---|---|---|---|---|
| | P@5 | P@10 | P@20 | R@5 | R@10 | R@20 | F1@5 | F1@10 | F1@20 | Jaccard |
| BPR | 0.4087 | 0.3418 | 0.2563 | 0.2066 | 0.3384 | 0.5004 | 0.2669 | 0.3298 | 0.3300 | - |
| GCN | 0.4765 | 0.3711 | 0.2792 | 0.2287 | 0.3536 | 0.5557 | 0.3017 | 0.3599 | 0.3613 | - |
| KGAT | 0.4832 | 0.3852 | 0.2956 | 0.2434 | 0.3835 | 0.5822 | 0.3152 | 0.3730 | 0.3812 | - |
| GAMENet | 0.5066 | 0.4176 | **0.3096** | 0.2557 | 0.4151 | **0.6027** | 0.3300 | 0.4037 | **0.3976** | 0.1874 |
| SafeDrug | 0.5038 | 0.4082 | 0.3000 | 0.2562 | 0.4105 | 0.5926 | 0.2672 | 0.3364 | 0.3534 | 0.1791 |
| SMGCN | 0.5248 | 0.4121 | 0.3027 | 0.2637 | 0.4136 | 0.5900 | 0.3380 | 0.3982 | 0.3887 | - |
| KDHR | 0.4329 | 0.3787 | 0.2872 | 0.2229 | 0.3862 | 0.5689 | 0.2680 | 0.3710 | 0.3715 | - |
| **Ours** | **0.5477** | **0.4275** | 0.3087 | **0.2727** | **0.4243** | 0.6010 | **0.3538** | **0.4128** | 0.3973 | **0.2447** |

Table 2: The difference and intersection herbs prescribed by our model and TCM doctor according to clinical symptoms and records of the same patient for two diagnoses.

| Sequential diagnoses | Symptom Set | Herb Set | |
|---|---|---|---|
| | | SCEIKG | TCM doctor |
| First diagnosis | 抑郁症 (depression)
口干 (xerostomia)
大便费力 (dyschezia)
入睡困难 (insomnia)
眠浅易醒 (light sleep, easy to wake up)
乏力 (fatigue)
胸闷 (chest tightness)
四肢麻木 (numbness of limbs)
舌淡红 (pale red tongue)
下睑淡白 (pale lower eyelid) | **黄芩 (scutellaria baicalensis)**
**炙甘草 (glycyrrhiza uralensis)**
**生姜 (ginger)**
**大枣 (jujube)**
**人参 (ginsen)**
桂枝 (cinnamomum cassia)
茯苓 (tuckahoe)
白芍 (paeonia lactiflora)
牡蛎 (ostrea gigas)
干姜 (zingiber officinale) | **黄芩 (scutellaria baicalensis)**
**炙甘草 (glycyrrhiza uralensis)**
**生姜 (ginger)**
**大枣 (jujube)**
**人参 (ginsen)**
北沙参 (radix adenophorae)
柴胡 (bupleuri radix)
天花粉 (flos rosae rugosae) |
| | **p@10=0.5000 r@10=0.6250 f1@10=0.5556** | | |
| Second diagnosis | 口干 (xerostomia)
惊恐 (panic)
焦虑 (anxiety)
入睡困难 (insomnia)
眠浅易醒 (light sleep, easy to wake up)
乏力 (fatigue)
胸闷 (chest tightness)
四肢麻木 (numbness of limbs)
小便频急 (frequent urination)
右手心热 (palm heat)
舌淡红 (pale red tongue)
苔薄 (thin fur)
下睑淡白边偏红 (pale lower eyelid with reddish edges) | **黄芩 (scutellaria baicalensis)**
**赤芍 (paeonia lactiflora)**
**炙甘草 (glycyrrhiza uralensis)**
**大枣 (jujube)**
**生姜 (ginger)**
**清半夏 (ternate pinellia)**
茯苓 (tuckahoe)
人参 (ginsen)
桂枝 (cinnamomum cassia)
炒六神曲 (medicated leaven) | **黄芩 (scutellaria baicalensis)**
**赤芍 (paeonia lactiflora)**
**炙甘草 (glycyrrhiza uralensis)**
**大枣 (jujube)**
**生姜 (ginger)**
**清半夏 (ternate pinellia)** |
| | **p@10=0.6000 r@10=1.0000 f1@10=0.7500** | | |

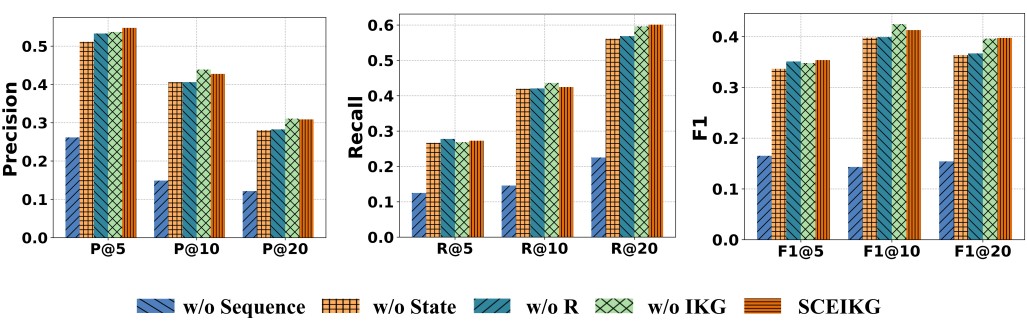

Figure 3: Performance of different variants of SCEIKG on different evaluation metrics.

Table 3: Performance of Different Knowledge Graph Completion Methods on ZzzTCM Dataset.

| Methods | Precision | | | Recall | | | F1 | | |
|---|---|---|---|---|---|---|---|---|---|
| | P@5 | P@10 | P@20 | R@5 | R@10 | R@20 | F1@5 | F1@10 | F1@20 |
| TransR Lin et al. (2015) | 0.4738 | 0.3658 | 0.2695 | 0.2517 | 0.3818 | 0.5475 | 0.3159 | 0.3602 | 0.3508 |
| TransE Bordes et al. (2013) | 0.5329 | 0.4315 | 0.3074 | 0.2680 | 0.4360 | 0.6077 | 0.3470 | 0.4211 | 0.3964 |
| ComplEx Trouillon et al. (2016) | 0.4966 | 0.4128 | 0.2936 | 0.2531 | 0.4258 | 0.5844 | 0.3260 | 0.4044 | 0.3798 |
| RotaoE Sun et al. (2019b) | 0.5114 | 0.3960 | 0.2889 | 0.2562 | 0.3903 | 0.5596 | 0.3316 | 0.3811 | 0.3713 |
| *rho*RotatE Sun et al. (2019b) | 0.5396 | 0.4396 | 0.3111 | 0.2687 | 0.4382 | 0.6131 | 0.3491 | 0.4261 | 0.4011 |
| DistMult Yang et al. (2014) | 0.4153 | 0.3980 | 0.2893 | 0.2490 | 0.4002 | 0.5673 | 0.3215 | 0.3866 | 0.3731 |
| **Ours** | **0.5477** | **0.4275** | **0.3087** | **0.2727** | **0.4243** | **0.6010** | **0.3538** | **0.4128** | **0.3973** |

## 4.1 COMPARISONS WITH BASELINES

We evaluate the performance of SCEIKG against several baseline models spanning different methods. As illustrated in Table 1, the traditional recommendation approach, **BPR** (Rendle et al., 2012),**GCN** (Kipf & Welling, 2016), **KGAT** (Wang et al., 2019a), **GAMNet** (Shang et al., 2019), **SafeDrug** (Yang et al., 2021), **SMGCN** (Jin et al., 2020), and **KDHR** (Yang et al., 2022). GAMENet and SafeDrug, primarily designed for Western drug recommendations and requiring additional ontology data, are not considered baselines. Also, when applying our dataset in KDHR, we omitted the herb knowledge graph module. In contrast, our model incorporates the condition changes, resulting in superior accuracy for TCM recommendations. Table 1 reveals SCEIKG's outperformance over GAMNet in *Top*-5 and *Top*-10 recommendations, with only a marginal difference in *Top*-20. However, in practical TCM recommendations, a smaller number of suggestions poses a more significant challenge. As a result, SCEIKG demonstrates significant advancements over baselines, showcasing its robust predictive power in herb prediction based on multiple patient diagnoses. Furthermore, we introduced the Jaccard metric to evaluate the set recommendation accuracy of our proposed method., we only compare the Jaccard similarity scores of the recommendations with those of two methods, SafeDrug and GAMENet, which are also sequence recommendation methods, and the results are shown also reflect the effectiveness of our model. Our experimental dataset, setting and specific parameter settings of baseline models are provided in Appendix A.

## 4.2 EXPERIMENTAL ANALYSIS

**Herb Compatibility** We present a heatmap of all herb pairs in Fig.4, with herb names omitted due to data privacy. From the heatmap, we observe that the magnified positions, specifically (Cassia Bark(肉桂)), Red Halloysite(赤石脂)) and (Danshen Root(丹参), Lightyellow Sophora Root(苦参)), both exhibit a correlation of 0. The correlation between "Cassia Bark(肉桂)" and "Red Halloysite(赤石脂)" is 0, indicating a certain degree of mutual antagonism between these two herbs. This aligns with the ancient literature's viewpoint that "Cassia Bark is effective in regulating cold energy, but it loses its efficacy when encountered with Red Halloysite(官桂善能调冷气，若逢石脂便相欺)." In other words, Cinnamon and Red Halloysite are mutually repellent. Additionally, "Danshen Root(丹参)" and "Lightyellow Sophora Root(苦参)" cannot be mixed in some situations due to their differing medicinal properties. **Nevertheless, real doctors also point out that some herb combinations may be controversial as they may have different effects in clinical practice. We acknowledge that we have not yet fully addressed the issue of herb interactions, but we have selected 70 recommended results for real doctors to analyze, and after real doctors' analysis,**

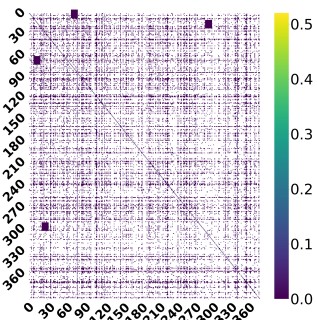

Figure 4: The visualization of the heatmap on the relationship between all herbs and a local zoom of incompatible herb pairs.

**we have achieved a 91.4% level of compatibility for our recommended herb pairings, which, in terms of our recommended results, demonstrates the validity of our method.**

**Herb Recommendations** We conduct a case study to verify the rationality of the herb recommendation for our proposed model. Table 2 shows examples in the herb recommendations scenario. Given the symptoms set for patient $x_j$, our proposed model generates an herb set to cure the listed symptoms. In the Herb Set column, the bold red font indicates the common herbs between the herb set recommended by our model and the herbs prescribed by TCM doctors. While our model also recommended some herbs not prescribed by the doctor and there are some discrepancies between the prescribed herbal prescriptions by SCEIKG and the actual prescriptions, their appropriateness for the symptom set has been verified by the TCM doctor. The initial diagnosis's prescription was considered by the doctors to be more applicable to the given symptom set than the real prescription, thus affirming the efficacy of our model. For the subsequent diagnosis, the recommended prescription showed no significant deviations from the real prescription, which was likewise deemed suitable for treating the symptom set.

**Evaluation of IKG** Due to the incorporation of the IKG embedding method in our herbal recommendation approach, we conducted experiments to further validate the impact of knowledge graph completion methods on our approach. We explored various knowledge graph completion methods, and the results in Table 3 demonstrate that different knowledge graph completion methods indeed have a certain influence on our herbal recommendation outcomes. Additionally, it can be observed that our approach, which combines TransR with RotatE, yielded the best results. This further underscores the effectiveness of our approach.

**Ablation Study** To further strengthen the credibility of our model, we conducted comprehensive comparisons with its variants to highlight the significance of each component. We introduced four model variants: (1) **SCEIKG w/o Sequence**: this variant applies the model without considering multiple diagnoses for sequential herb recommendation and the transition condition module. (2) **SCEIKG w/o State**: this variant does not take into account constraints in the patient's condition (denoted as $L_{state}$). (3) **SCEIKG w/o $R$**: this variant excludes herbal compatibility constraints (denoted as $R$). (4) **SCEIKG w/o IKG**: this variant is based on the initial model but excludes TransR combined with the RotatE embedding component and the correlation attention mechanism, and we train the model without $L_{IKG}$. As shown in Fig.3, which verifies the importance of each component of the model, the reason for the very poor performance of SCEIKG without sequence is that it relies on whether sequential information and transition conditions involve the model or not. We observed that considering the changes in the patient's condition after taking the herbs significantly improves the predictive performance. Also, we have analyzed the reasons for the poor effectiveness without sequence in Appendix D and the slight decrease in some metrics for SCEIKG compared to SCEIKG without IKG in Appendix E.

## 5 CONCLUSION

Our paper claims that to cope with an accurate herb recommendation, learning must take into account how internal changes in taking medication for patients. Toward this, we investigate TCM recommendation tasks from novel perspective of incorporating the sequential diagnoses for patients and develop a condition module to simultaneously learn the condition embedding and guide the next diagnosis. In the experiment, the dataset is mainly composed of insomnia cases. The radar chart in Fig.5 reveals that the top ten symptoms and herbs are closely related to insomnia, indicating a limitation in covering diverse types of cases. Also, our modeling of patient state transitions is implicit. In the future, it would be interesting to enhance the model's robustness by explicitly representing patient state transitions and addressing these limitations by integrating specific knowledge from the field of traditional Chinese medicine, including dosage information and contraindications, within the interactive knowledge graph. Furthermore, due to he unique diagnostic and treatment methods in TCM, characterized by high personalization and holism, differ significantly from conventional medicine. TCM often involves complex herbal combinations tailored to the overall condition of the patient. In contrast, general medicine tends to rely on quantifiable indicators, such as data from medical instruments. While our approach may be applicable to other forms of medicine, its design is better suited for TCM recommendations, focusing on domain knowledge rooted in TCM principles.

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

## A ADDITIONAL EXPERIMENT INFORMATION

### A.1 ZZZTCM DATASET AND IKG PROCESSING

**ZzzTCM Dataset** To guarantee the authenticity of the TCM recommendation data, we curated a new sequential real-world dataset that includes patients' multiple diagnoses, named ZzzTCM (a play on "Zzz" for sleep and TCM for traditional Chinese medicine). We source the medical records from Guangdong Provincial Hospital of Traditional Chinese Medicine, which covers patient information nationwide rather than being confined to Guangdong Province. The hospital had already sought the consent of the related patients to use their medical history for academic research. Unlike data directly crawled from online communities, these hospital records represent actual cases diagnosed by medical professionals, ensuring higher quality. A total of 17,000 historical records were provided, from which we selected 751 patient records and corresponding TCM prescriptions from practitioners. Each patient underwent $1 \sim 17$ multiple follow-up diagnoses. We extracted patient narratives from the dataset provided by the Provincial Traditional Chinese Medicine (TCM) institution, focusing on symptoms related to insomnia. By merging records with the same patient EMPI and diagnosis ID, we obtained the medical histories of patients who had multiple diagnoses. During data preprocessing, we filtered out blank medical records and used the ChatGPT API called the gpt-3.5-turbo model to extract the patients' symptoms set based on the prompt "Please extract the keywords of the patient's relevant symptoms". After that, we consolidated data from all diagnoses of the same patient and transformed symptoms and herbs into multi-hot vectors before training.

**Interaction Knowledge Graph** We construct an Interaction Knowledge Graph (IKG) from multiple data sources. The IKG contains $\mathcal{R}$ edge relations and $V$ entities, which include bidirectional edge directions, such as symptom-related herbs, and herb-related symptoms. The entities of IKG contain herbs, symptoms, diseases, pathogeny, et al. At the same time, we indexed the herbs and symptoms, starting from 0, and constructed triples based on the co-occurrence of symptoms and herbs, which are added to the constructed knowledge graph to form an Interaction Knowledge Graph (IKG). If an entity in the constructed initial knowledge graph was not found in the herbs or symptoms, we continued indexing to ensure that all entities in the IKG had index values. We further trained the IKG using completion embedding techniques.

The process of initial knowledge graph construction was divided into two main stages: first, we dynamically acquired data from websites in the relevant domains using crawling techniques such as Python, the relevant URLs are labeled in Table 4, which were subsequently systematically cleaned and organized. Next, the second step is to extract relevant ternary knowledge from the web data by applying manually designed rules. For example, we can convert information such as "Yang Er Ju has the efficacy of moving qi and relieving pain, and can treat wind-heat and colds" into the representation of TCM knowledge such as (Yang Er Ju, drug main treatment, wind-heat and colds) and (Yang Er Ju, drug-related effects, moving qi and relieving pain). Through these steps, we constructed the initial knowledge graph.

The statistics of the ZzzTCM dataset and IKG are reported in Table 4. In addition, common symptoms and herbal statistics are shown in Fig.5 and part of the IKG is shown in Fig.6.

### A.2 METRICS DETAILS

Given a symptom set $\mathbb{S}$ and record $r$, our proposed model generates a herb set $\hat{Y}$. To evaluate the performance of *Top-K* recommended herbs, we adopt three evaluation metrics: Precision@*Top-K*, Recall@*Top-K*, and F1@*Top-K*. The *Prescision* score indicates the hit ratio of herbs is true herbs. And the *Recall* describes the coverage of true herbs as a result of a recommendation. $F_1$ is the harmonic mean of precision and recall. In particular, we obtain the evaluation score in the test data by taking the average of patients' diagnoses.

$$Precision_j^{(t)} = \frac{1}{X}\frac{1}{T}\sum_{j=1}^{X}\sum_{t=1}^{T}\frac{|\{i : hc_{j,i}^{(t)} = 1\} \cap \{i : \text{Top}(\hat{Y}_{j,i}^{(t)})\}|}{|\{i : \text{Top}(\hat{Y}_{j,i}^{(t)})\}|} \tag{7}$$

$$Recall_j^{(t)} = \frac{1}{X}\frac{1}{T}\sum_{j=1}^{X}\sum_{t=1}^{T}\frac{|\{i : hc_{j,i}^{(t)} = 1\} \cap \{i : \text{Top}(\hat{Y}_{j,i}^{(t)})\}|}{|\{i : hc_{j,i}^{(t)} = 1\}|} \tag{8}$$

Table 4: Data Statistics

| | Items | Size |
|---|---|---|
| ZzzTCM | # of diagnoses / # of patients | 2761/751 |
| | symptoms. / herbs. space size | 6562/387 |
| | avg. / max # of diagnoses | 3.68/17 |
| IKG | entities | 344092 |
| | relations | 35 |
| | triples | 4308799 |
| | data source | TCM (Yao et al., 2018) |
| | | ZzzTCM |
| | | Chinese Medicine Knowledge Base Website [a] |
| | | Chinese Medicine Family Website [b] |
| | | Seeking Medicine Help Website [c] |

[a] http://tcm.med.wangfangdata.com.cn
[b] http://tcm.med.wangfangdata.com.cn
[c] https://www.zysi.com.cn/zhongyaocai/index.html

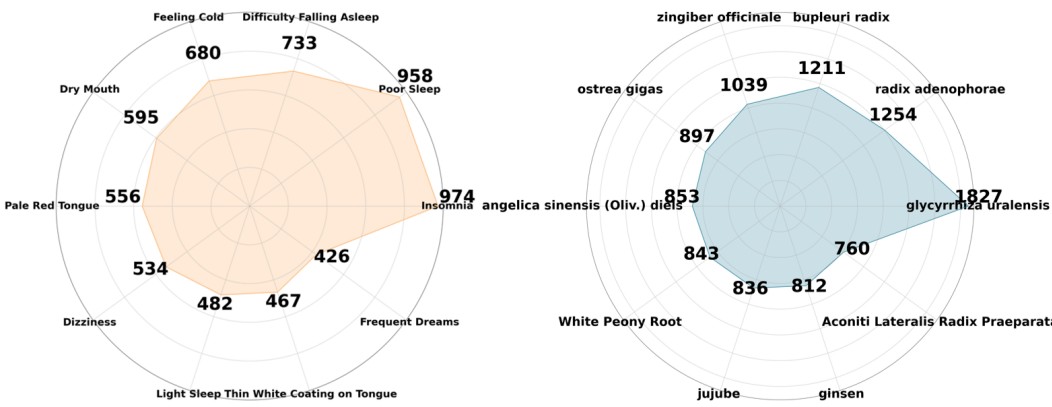

Figure 5: Visualization of Common Symptoms and Herbs Radar Chart, with orange section showing common symptoms and blue section displaying common herbs.

$$\text{F1}_j^{(t)} = \frac{2}{\frac{1}{\text{Precision}_j^{(t)}} + \frac{1}{\text{Recall }_j^{(t)}}}. \tag{9}$$

where $hc_j^{(t)}$ is the ground-truth herb prescription during the $t$-th diagnosis of patient $j$, and $hc_{j,i}^{(t)}$ is the $i$-th element. $\text{Top}(\hat{Y}_{j,i}^{(t)})$ is the top $i$-th element with the highest prediction scores. $|\cdot|$ denotes the cardinality, $\cap$ is set interaction operation. The *Prescision* score indicates the hit ratio of herbs is true herbs. And the *Recall* describes the coverage of true herbs as a result of a recommendation. $F_1$ is the harmonic mean of precision and recall. In particular, we obtain the evaluation score in the test data by taking the average of patients' diagnoses.

## A.3 EXPERIMENTAL SETTINGS

In this paper, we ran all experiments on a platform with Ubuntu 16.04 on 256GB of memory and an NVIDIA GeForce GTX 1080 Ti GPU. And we implement our model in PyTorch. In the training, we used a random seed of fixed size 2019 to guarantee the reproducibility of the results. The overall

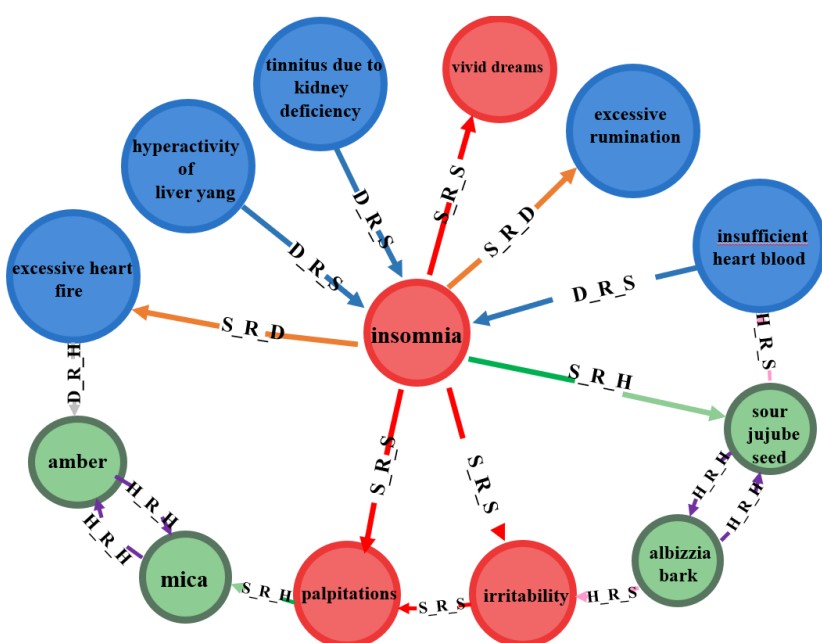

Figure 6: Local visualization of the Interaction Knowledge Graph $\mathcal{G}$. D, S, H, and R represent the disease, symptom, herb, and relationship, respectively.

framework was optimized with Adam optimizer, where the batch size is one patient with all diagnoses and the batch size of IKG is fixed at 2048. The length $\Gamma = 128$ of the record sequence $r^{(t)}$. We set the multi-hop $k$ of GNN with Hierarchical Attention-based UPDATE to three with hidden dimensions $D_k = [64, 32, 16]$, in order to model the third-order connectivity; The embedding size of entity $e_h$ and $e_r$ is fixed to 64. In the transition condition module, we set the hidden size $hidden\_dim = 64$ of LSTM. And we trained the model for 1000 epochs with a learning rate $lr = 0.0001$ and the coefficient of normalization $\lambda_\Theta = 10^{-5}$. For evaluation metrics, we set $Top\text{-}K = [5, 10, 20]$. We report the average metrics for all patients in the test set. Moreover, an early stopping strategy is suggested (Wang et al., 2019a), premature stopping if $recall@Top\text{-}K = 20$ on the validation set does not increase for $early\_stop = 50$ successive epochs. The default Xavier initializer (Glorot & Bengio, 2010) to initialize the model parameters. Also, we conduct experiments on parameter sensitivity, which are presented in Appendix C.

## A.4 BASELINES DETAILS

In this paper, we evaluate the performance of SCEIKG by comparing it against the following baselines. To carry out a fair comparison, all experiments are run on the same platform. We also utilize 64 batch sizes for traditional recommendation approaches and one patient for sequence-based models. Also, the early stop mechanism is also applied in the baseline methods and the number of layers is set to 3 for GCN-based models.

**BPR** (Rendle et al., 2012) performs poorly due to its neglect of multi-hop interactions and the evolving nature of a patient's condition. It presents a generic optimization criterion BPR-Opt for personalized ranking which is the maximum posterior estimator derived from a Bayesian analysis of the traditional recommendation. 64-dim embedding tables implement the model, the learning rate $10^{-3}$, 64 batch size, and the Adam as optimizer. Also, we utilize an early stop mechanism to train all models.

**GCN** (Kipf & Welling, 2016) introduces the degree matrix of the node to solve the problem of self-loops and the normalization of the adjacency matrix and sums the embedding of neighbor nodes to update the current node. The parameter settings of GCN are the same as the SMGCN (Jin et al., 2020).

**KGAT** (Wang et al., 2019a) incorporates higher-order collaborative signals for traditional recommendations, it falls short in exploring the higher-order relationships specific to TCM recommendations, namely the connections between symptom sets and herb sets. Following the original paper, we implement the 64-dim embedding tables. Adam is used as the optimizer with a learning rate at $10^{-4}$.

**GAMENet** (Shang et al., 2019) and **SafeDrug** (Yang et al., 2021) capture comprehensive medical histories of patients utilizing longitudinal vectors of medical codes. These models solely consider the patient's medication records and fail to capture the nuanced aspects of the physique. Although GAMENet exhibits some similarities to our model when $Top$-$K$ is set to 20, its performance lags behind ours for other $Top$-$K$ values, thus underscoring the strength of our patient history-based approach. We use the same suit of hyperparameters reported in the original papers: the learning rate at $5 \times 10^{-4}$ use 64-dim embedding tables and 64-dim GRU as RNN.

**SMGCN** (Jin et al., 2020) obtains the embedding of symptoms and herbs and recommends herbs through an implicit syndrome induction process. For SMGCN, the learning rate is $2e - 4$ and the dimension of the GCN layer is 128. The regularization coefficient is set to $7 \times 10^{-3}$, the dimensions of the embedded layer and the hidden layer are 64, the GCN output dimension of the last layer is 256 and the MLP layer size is 256.

**KDHR** (Yang et al., 2022) introduces herb properties as additional auxiliary information by constructing an herb knowledge graph and employs a graph convolution model with multi-layer information fusion to obtain symptom and herb feature representations. the initial learning rate is $3 \times 10^{-4}$, Adam is used to optimize the parameters, and the regularization coefficient is set to 0.007.

Although **SMGCN** (Jin et al., 2020) and **KDHR** (Yang et al., 2022) achieve excellent accuracy for TCM recommendation, both models neglect the crucial aspect of accounting for changes in a patient's condition over time. Note that GAMENet and SafeDrug are not considered baseline since they require extra ontology data. Also, when applying our dataset in KDHR, we removed the module for the herb knowledge graph.

# B  Training Algorithm for SCEIKG model and Inference

Table 5: Hyperparameter experiment results.

| Hyperparameters | | Precision | | | Recall | | | F1 | | |
|---|---|---|---|---|---|---|---|---|---|---|
| | | P@5 | P@10 | P@20 | R@5 | R@10 | R@20 | F1@5 | F1@10 | F1@20 |
| $lr$ | 0.01 | 0.1664 | 0.0089 | 0.0758 | 0.0885 | 0.0938 | 0.1544 | 0.1096 | 0.0872 | 0.0984 |
| | 0.001 | 0.4322 | 0.3490 | 0.2638 | 0.1953 | 0.3227 | 0.5000 | 0.2633 | 0.3249 | 0.3361 |
| | 0.0001* | **0.5477** | **0.4275** | **0.3087** | 0.2727 | 0.4243 | **0.6010** | 0.3538 | 0.4128 | **0.3973** |
| | 0.00001 | 0.5383 | 0.4248 | 0.2990 | **0.2807** | **0.4318** | 0.5947 | **0.3550** | **0.4130** | 0.3867 |
| $\lambda_\Theta$ | $1.0 \times 10^{-4}$ | 0.5302 | 0.4101 | **0.2883** | 0.2688 | 0.4091 | **0.5655** | 0.3466 | 0.3973 | **0.3721** |
| | $1.0 \times 10^{-5}$ * | **0.5477** | **0.4275** | **0.3087** | **0.2727** | 0.4243 | **0.6010** | **0.3538** | 0.4128 | **0.3973** |
| | $1.0 \times 10^{-6}$ | 0.5436 | 0.4349 | 0.3027 | 0.2731 | 0.4337 | 0.5902 | 0.3534 | 0.4209 | 0.3896 |
| $\Gamma$ | 32 | 0.5409 | 0.4376 | 0.3047 | 0.2724 | **0.4355** | 0.5914 | 0.3521 | **0.4232** | 0.3916 |
| | 64 | 0.5208 | 0.4060 | 0.2849 | 0.2654 | 0.4200 | 0.5748 | 0.3418 | 0.3982 | 0.3700 |
| | 128* | **0.5477** | **0.4275** | **0.3087** | **0.2727** | 0.4243 | **0.6010** | **0.3538** | 0.4128 | **0.3973** |
| $hidden\_dim$ | 32 | 0.5423 | **0.4403** | 0.3007 | 0.2692 | **0.4314** | 0.5829 | 0.3496 | **0.4225** | 0.3864 |
| | 64* | **0.5477** | 0.4275 | **0.3087** | **0.2727** | 0.4243 | **0.6010** | **0.3538** | 0.4128 | **0.3973** |
| | 128 | 0.5289 | 0.4262 | 0.3070 | 0.2687 | 0.4281 | 0.5979 | 0.3464 | 0.4142 | 0.3954 |
| | 256 | 0.5289 | 0.4208 | 0.3007 | 0.2682 | 0.4261 | 0.5940 | 0.3455 | 0.4083 | 0.3881 |

\* Asterisks indicate baseline experiment settings

**Algorithm** We provide further insights into the implementation of our Algorithm 1, which is rooted in the Expectation Maximization (EM) algorithm (Dempster et al., 1977), a well-established iterative optimization strategy. Our model is built upon a similar conceptual framework as the EM algorithm. Initially, we engage in complementary learning of the knowledge graph, involving updates to the Interaction Knowledge Graph (IKG) to enhance the embedded representations of entities. Subsequently, these enriched entity embeddings are applied in the training phase for TCM recom-

---

**Algorithm 1** Training of SCEIKG

---

**Require:** Training set $\iota$, Interaction knowledge graph $\mathcal{G}$, weight matrix of IKG $\mathbb{W}$ in Eq.(1), batch of patients $\zeta$, batch of Triplets $\xi$, total number of patients $\eta$, total number of epoch $E$, the configuration $\Theta$

**Output: herb set** $\hat{Y}$

Initialize all configurations $\Theta$

**for** epoch $\leftarrow 0, 1, \cdots, E$ **do**

    Generate Entities Embedding $E \in \mathbb{R}^{V \times D_{out}}$, propagate over the interaction knowledge graph

    /*Phase I: Interaction Knowledge graph Complementation*/

    **for** triples $(h, r, t)$ in $\mathcal{G}$ of batch $\xi$ **do**

        Calculate the score of the knowledge triples $f(h, r, t)$

        Calculate the interaction knowledge graph loss $L_{IKG}$ and update interaction knowledge graph embedding $e_h$.

    **end for**

    Update the weight matrix $\mathbb{W}$ of $\mathcal{G}$ by the function UPDATE$(\cdot)$

    /*Phase II: Recommended herbs based on sequential diagnoses for each patient*/

    **for** batch j:=1 to $\frac{\eta}{\zeta}$ **do**

        Select the batch of patients sequential records $\Omega$

        /*Note that the current diagnosis contains $\xi$ patients*/

        **for** diagnosis $t := 1$ to $|T|$ **do**

            **if** t==1 **then**

                Select the $t$-th batch of patient, $\Omega^{(t)}$

            **else**

                Select the $t$-th batch of patient, $\Omega^{(t)}$ and Transition condition representation $\Psi^{(t-1)}$

            **end if**

            Generate Condition Representation $h_{\mathbb{C}}^{(t)}$ and Symptom Representation $h_{\mathbb{S}}^{(t)}$

            /*The first diagnosis is not having the previous patient's condition and the last diagnosis is not having the next condition*/

            Generate Horizontal Patient Representation $\Phi_P^{(t)}$ based on the Transition Condition Module $\Psi^{(t-1)}$

            Generate Patient to herb Matching $\hat{Y}^{(t)}$

        **end for**

        Accumulate the loss of herb prediction and update the configuration $\Theta$ by Adam

    **end for**

**end for**

---

Table 6: Performance Results of Different Partition Ratios for the ZzzTCM Training Set.

| | Precision | | | Recall | | | F1 | | |
|---|---|---|---|---|---|---|---|---|---|
| | P@5 | P@10 | P@20 | R@5 | R@10 | R@20 | F1@5 | F1@10 | F1@20 |
| 0.2 | 0.4813 | 0.3919 | 0.2809 | 0.2454 | 0.3913 | 0.5514 | 0.3164 | 0.3803 | 0.3627 |
| 0.4 | 0.4648 | 0.3847 | 0.2912 | 0.2352 | 0.3864 | 0.5722 | 0.3041 | 0.3741 | 0.3758 |
| 0.6 | 0.4933 | 0.3960 | 0.2951 | 0.2571 | 0.4088 | 0.5933 | 0.3287 | 0.3901 | 0.3835 |
| 0.8 | 0.4959 | 0.4025 | 0.2879 | 0.2551 | 0.4089 | 0.5733 | 0.3264 | 0.3931 | 0.3867 |
| 0.9 | 0.5084 | 0.4206 | 0.3040 | 0.2470 | 0.4021 | 0.5718 | 0.3235 | 0.3994 | 0.3758 |
| 0.94 | **0.5477** | **0.4275** | **0.3087** | **0.2727** | **0.4243** | **0.6010** | **0.3538** | **0.4128** | **0.3973** |
| 0.98 | 0.6000 | 0.4579 | 0.3066 | 0.3059 | 0.4613 | 0.6151 | 0.3932 | 0.4465 | 0.3987 |

mendations. These two components of the cycle iteratively interact until the model converges, and the optimization process concludes.

As illustrated in Fig.2, the IKG updates play a pivotal role in refining the individual modules depicted in the figure. Conversely, the training of the model reciprocally enhances the IKG, as shown on the right. This dynamic interaction fosters iterative improvement. It is well-recognized that constructing a comprehensive knowledge graph for TCM is an intricate task that necessitates extensive data support. Therefore, knowledge graph complementation, which involves inferring new informa-

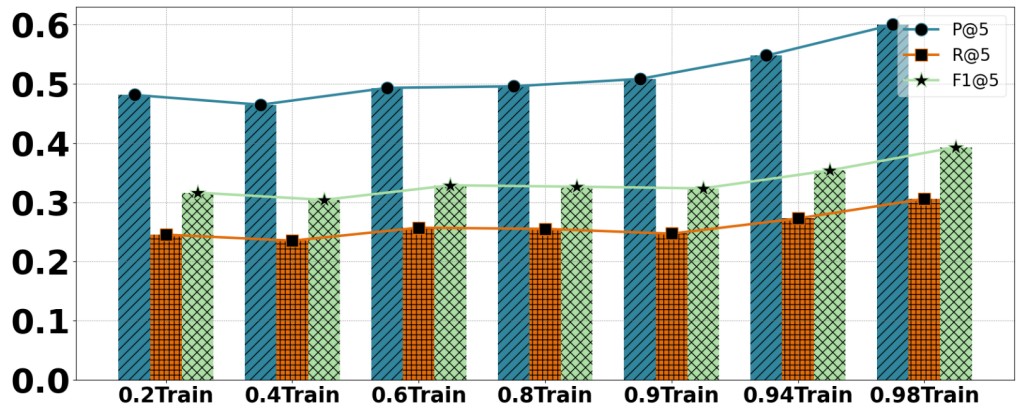

Figure 7: **Performance trend chart with different training set split ratios under Top@5.**

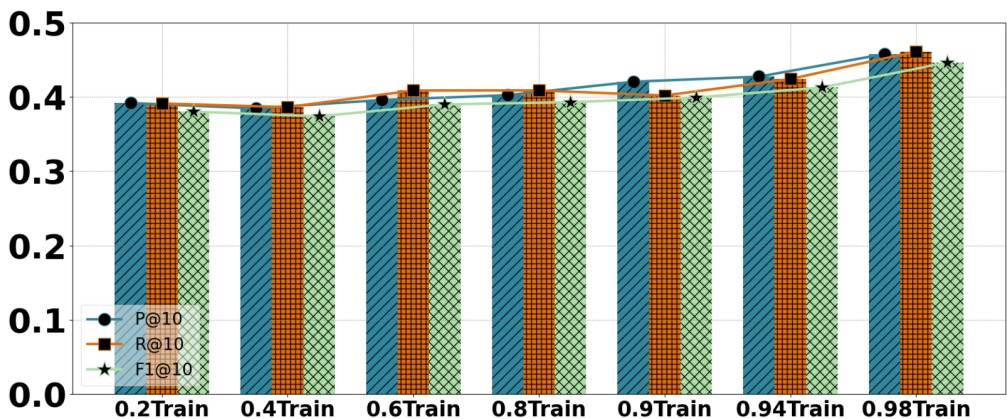

Figure 8: **Performance trend chart with different training set split ratios under Top@10.**

tion from existing data. In the first phase of Algorithm 1 (**Phase I: Interaction Knowledge graph Complementation**). The representation entities in the IKG are learned by Eq.3 and then updated inversely by $L_{IKG}$ in the loss function in Eq.6, which complements and enriches the information of the entities. The backpropagation process of the first phase is shown,

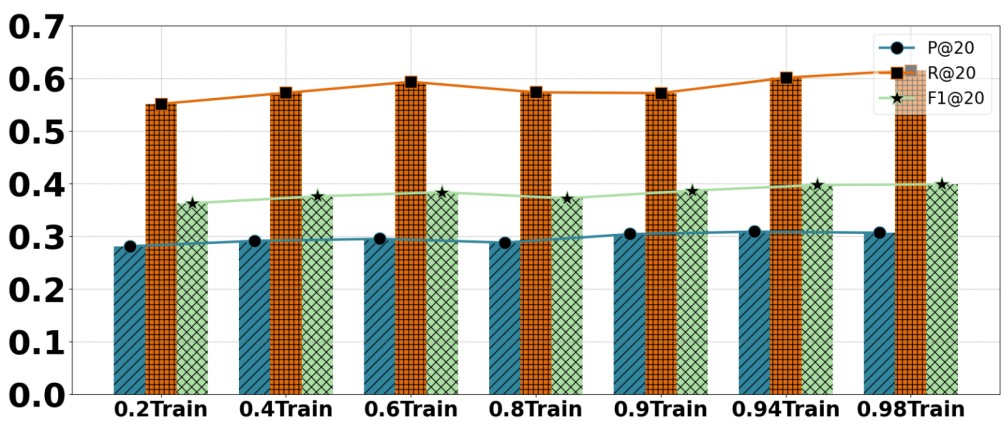

Figure 9: **Performance trend chart with different training set split ratios under Top@20.**

$$\frac{\partial \mathcal{L}_{IKG}}{\partial \boldsymbol{f}\left(h,r,t'\right)} = -\frac{1}{\sigma\left(\boldsymbol{f}\left(h,r,t'\right) - \boldsymbol{f}(h,r,t)\right)} \cdot \frac{d}{dx}\sigma\left(\boldsymbol{f}\left(h,r,t'\right) - \boldsymbol{f}(h,r,t)\right)$$
$$= -\frac{1}{\sigma\left(\boldsymbol{f}\left(h,r,t'\right) - \boldsymbol{f}(h,r,t)\right)} \cdot \sigma\left(\boldsymbol{f}\left(h,r,t'\right) - \boldsymbol{f}(h,r,t)\right) \cdot \left(1 - \sigma\left(\boldsymbol{f}\left(h,r,t'\right) - \boldsymbol{f}(h,r,t)\right)\right)$$
$$= -\left(1 - \sigma\left(\boldsymbol{f}\left(h,r,t'\right) - \boldsymbol{f}(h,r,t)\right)\right)$$

$$(10)$$

$$\frac{\partial \mathcal{L}_{IKG}}{\partial \boldsymbol{f}(h,r,t)} = \frac{1}{\sigma\left(\boldsymbol{f}\left(h,r,t'\right) - \boldsymbol{f}(h,r,t)\right)} \cdot \frac{d}{dx}\sigma\left(\boldsymbol{f}\left(h,r,t'\right) - \boldsymbol{f}(h,r,t)\right)$$
$$= \frac{1}{\sigma\left(\boldsymbol{f}\left(h,r,t'\right) - \boldsymbol{f}(h,r,t)\right)} \cdot \sigma\left(\boldsymbol{f}\left(h,r,t'\right) - \boldsymbol{f}(h,r,t)\right) \cdot \left(1 - \sigma\left(\boldsymbol{f}\left(h,r,t'\right) - \boldsymbol{f}(h,r,t)\right)\right)$$
$$= 1 - \sigma\left(\boldsymbol{f}\left(h,r,t'\right) - \boldsymbol{f}(h,r,t)\right)$$

$$(11)$$

the gradient descent derivation for $L_{IKG} = \sum_{(h,r,t,t')\in\mathcal{T}} -ln\sigma\left(\boldsymbol{f}\left(h,r,t'\right) - \boldsymbol{f}(h,r,t)\right)$. Then, we compute the gradient of the loss with respect to $\boldsymbol{f}\left(h,r,t'\right)$. Similarly, compute the gradient of the loss with respect to $\boldsymbol{f}\left(h,r,t\right)$. Then, compute the gradients of $\boldsymbol{f}\left(h,r,t'\right)$ and $\boldsymbol{f}(h,r,t)$ with respect to their respective embeddings:

$$\frac{\partial \boldsymbol{f}\left(h,r,t'\right)}{\partial h} = \frac{\partial}{\partial h}\left(C \cdot \|\sin\left(W_r e_h + e_r - W_r e_{t'}\right)\|_1\right) = C\frac{\partial}{\partial h}\|\sin\left(W_r e_h + e_r - W_r e_{t'}\right)\|_1 \quad (12)$$

$$\frac{\partial \boldsymbol{f}(h,r,t)}{\partial h} = \frac{\partial}{\partial h}\left(C \cdot \|\sin\left(W_r e_h + e_r - W_r e_t\right)\|_1\right) = C\frac{\partial}{\partial h}\|\sin\left(W_r e_h + e_r - W_r e_t\right)\|_1 \quad (13)$$

In gradient descent, update the parameters in the opposite direction of the gradient to minimize the loss function. Assuming that the parameters influencing $f(h,r,t)$ are denoted by $\theta$. The update rule for the parameters would be,

$$\theta \leftarrow \theta - \eta \cdot \frac{\partial L_{IKG}}{\partial \boldsymbol{f}(h,r,t)} \cdot \frac{\partial \boldsymbol{f}(h,r,t)}{\partial \theta} \quad (14)$$

Where $\eta$ is the learning rate, $\frac{\partial L_{IKG}}{\partial \boldsymbol{f}(h,r,t)}$ is the gradient we calculated, $\frac{\partial \boldsymbol{f}(h,r,t)}{\partial \theta}$ is the gradient of the score function with respect to the parameters $\theta$. Here, we provide an approximate derivation of the derivation descent derivation equation for $L_{IKG}$.

In traditional knowledge graph embedding methods only individual knowledge triples can be represented efficiently, and higher-order similarities between entities cannot be captured. However, this higher-order information is critical for understanding complex interactions between entities, especially in the field of herbal medicine recommendation. We detail how the entity representations obtained in Phase I can be applied to the recommendation of herbs in Phase II **(Phase II: Recommended herbs based on sequential diagnoses for each patient)** using a Graph Neural Network (GNN) approach. In this process, we use the adjacency matrix, which is constructed by the normalized pointwise mutual information method while utilizing the entity knowledge representations learned in the first stage. Then, we use the following matrix according to Eq.2 to aggregate higher-order relationships, as well as similarity information between higher-order entities. In addition, we introduce an attention mechanism for entity correlation, which is used to update the structure of the IKG graph, thus further enriching the information on herbal recommendations. Ultimately, we back-propagate through the loss function in Eq.6. This loss function includes a mean square error loss $L_{mse}$ for measuring the distance between the recommended set of herbs and the actual set of herbs, and a state loss $L_{state}$ for measuring the similarity of the states before and after the recommendation of the herbs as well as a regularization term $R$ for placing constraints on the relationships between herbs. The specific back propagation derivation is shown below,

$$\frac{\partial L_{mse}}{\partial \Theta} = 2\sum_{i=1}^{N}\left(hc_i^{(t)} - \hat{Y}_i^{(t)}\right) \cdot \frac{\partial \hat{Y}_i^{(t)}}{\partial \Theta} \quad (15)$$

Then, calculate the gradient of $L_{\text{state}}$ with respect to the parameter $\Theta$,

$$\frac{\partial L_{state}}{\partial \Theta} = \frac{\partial}{\partial \Theta} \left( \frac{h_{\mathbb{C}}^{(t+1)} \cdot \Psi^{(t)}}{\left\| h_{\mathbb{C}}^{(t+1)} \right\| \cdot \left\| \Psi^{(t)} \right\|} \right) \tag{16}$$

Since $h_{\mathbb{C}}^{(t+1)}$ and $\Psi^{(t)}$ have nothing to do with the parameter $\Theta$, we only need to calculate the gradient of the numerator part. Finally, the gradient of the regularisation term $R$ with respect to the parameter $\Theta$ is calculated as follows.

$$\frac{\partial R}{\partial \Theta} = -\sum_{i=1}^{N} \sum_{j=1}^{N} \mathbb{W}_{ij}^{h} \hat{Y}_i^{(t)} \cdot \frac{\partial \hat{Y}_i^{(t)}}{\partial \Theta} \cdot \hat{Y}_j^{(t)} - W_{ij}^{h} \hat{Y}_i^{(t)} \cdot \hat{Y}_j^{(t)} \cdot \frac{\partial \hat{Y}_j^{(t)}}{\partial \Theta} \tag{17}$$

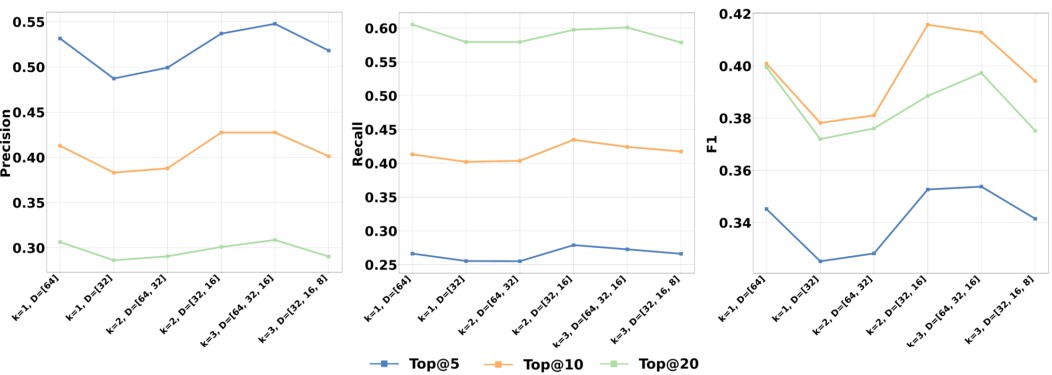

Figure 10: Effect of dimensions with different layers on SCEIKG.

Combining the above three components, the individual gradients are summed up to give the total gradient $\frac{\partial}{\partial \Theta} \left( L_{mse} + L_{state} + \lambda R \right)$. This total gradient will be used in the gradient descent optimization algorithm to update the parameter $\Theta$ to minimize the overall loss function. Ultimately, by continually iterating this gradient descent process, we can optimize the parameters of the model, thus minimizing the overall loss function and achieving the optimization goal of our model. These two stages iteratively update each other and finally complete the training of the whole model.

Note that for each patient at the first diagnosis, we do not have access to the patient's previous state and therefore cannot perform state transfer prediction. Only after the first diagnosis can we start modeling the patient's historical state dynamically. At the last diagnosis, we did not acquire the patient's next state either, so we need to pay attention to the boundary condition handling in modeling. In addition, there are different numbers of diagnoses for each patient, so we first pad the batch of patients to the same number of diagnoses, but during training, the padding data is not entered into forward propagation. Hence, we can recommend multiple patients in parallel.

**Training Inference** The model is trained end-to-end. We optimize the prediction loss and $L_{IKG}$ alternatively. In particular, for a batch of randomly sampled $(h, r, t, t')$, we update the embeddings for all nodes; hereafter, we sample a batch of patients with consecutive diagnoses randomly, retrieve their representations after $L$ steps of propagation, and then update model parameters by utilizing the gradients of the prediction loss. Finally, we select the top $K$ herbs with the highest probabilities as the recommended herb set.

## C  PARAMETER SENSITIVITY

In this section, we apply a grid search for hyper-parameters: the learning rate is tuned amongst $lr = \{0.01, 0.001, 0.0001, 0.00001\}$, the coefficient of normalization $\lambda_{\Theta}$ is searched in $\{10^{-4}, 10^{-5}, 10^{-6}\}$. We tune the max length $\Gamma = \{32, 64, 128\}$ of the condition to explore the

impact of changes in historical medication patient status. The hidden dimensional size of LSTM $hidden\_dim = \{32, 64, 128\}$ to capture the useful information across multiple diagnoses. Hyperparameter experiment results are provided in Table 5. The model tends to be robust to hyperparameter changes. Also, we explore whether our proposed model can benefit from a larger number of embedding propagation layers, we tune the depth of GNN layers on the submodel, which is varied in $k = \{1, 2, 3\}$ combined with the different dimensions $kd$ of each layer. The result is shown in Fig.10. Intuitively, this is because more vectors can encode more useful information in latent space. **However, due to the limitations of large knowledge graphs received from experimental conditions, we are not able to conduct higher dimensional exploration.**

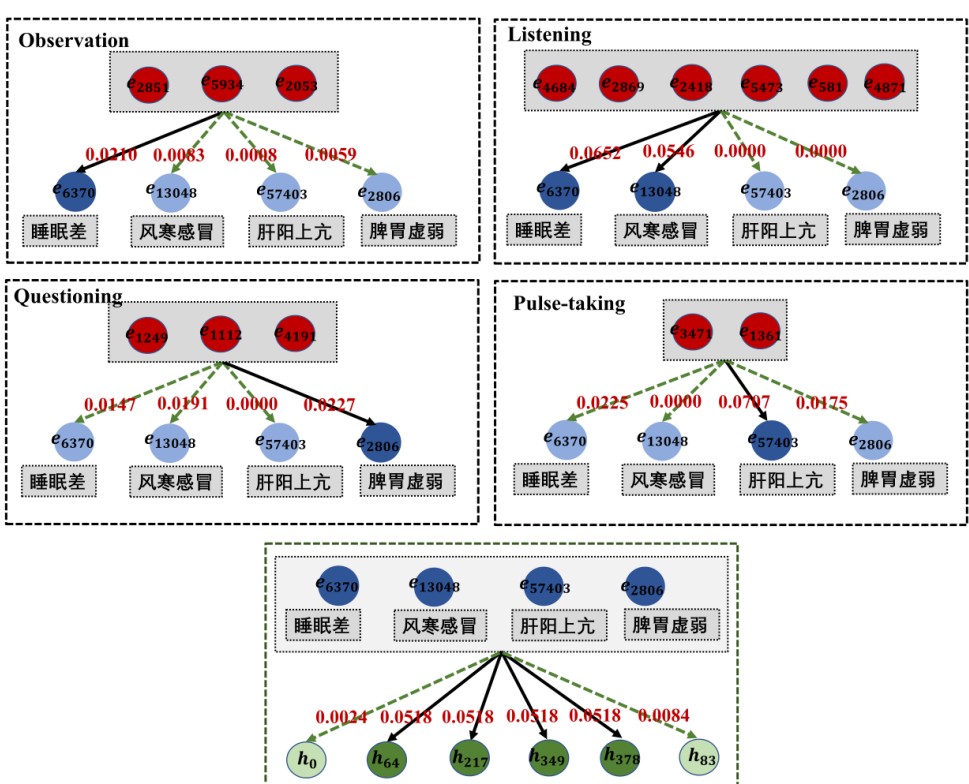

Figure 11: Real example of IKG aggregation in SCEIKG. The process of observation of the symptoms entity $\{e_{2851}, e_{5934}, e_{2053}\}$ corresponds respectively to { pale redness in the lower eyelid, white and greasy coating on the tongue, dark red tongue }; the process of Listening of the symptoms entity $\{e_{4684}, e_{2869}, e_{2418}, e_{5473}, e_{581}, e_{4871}\}$ corresponds respectively to { dry mouth without bitterness, burning sensation in the eye corners, tidal heat, sneezing, migraine }; the process of questioning of the symptoms entity $\{e_{1249}, e_{1112}, e_{4191}\}$ corresponds respectively to {borborygmi, abdominal distension, poor appetite }; the process of pluse-taking of the symptoms entity $\{e_{1249}, e_{1112}, e_{4191}\}$ corresponds respectively to {ulnar pulse (TCM) and pulse string-like taut}.

Furthermore, due to the relatively small size of our dataset, we also performed different ratio splits on the ZzzTCM training data. Table 6 presents the evaluation metric results for different split ratios. To provide a more intuitive view of the results, Fig.7, Fig.8 and Fig.9 respectively show the trend charts for different training set ratios under the same Top@k metric. From the results, it can be observed that as the training set size increases, the evaluation metrics show an upward trend to some extent. This indicates that dataset size is not the sole determining factor for model performance, further evaluating the robustness of our model.

## D  ADDITIONAL EXPERIMENTS

**Interpretability of Recommendation** To further explain our recommendation results and the interpretability of our model, we employ high-order connectivity reasoning to infer the prescription for the current condition of the patient. It is noteworthy that, given the symptoms and herbs are both in set forms, high-order connectivity is based on the weight matrix values of the IKG. We choose neighboring entities with larger adjacent weights for information aggregation, rather than simply displaying the path selection. We randomly selected records of a patient from the ZzzTCM dataset, and due to privacy concerns, we briefly introduce the symptoms: pale redness in the lower eyelid, white and greasy coating on the tongue, dark red tongue, dry mouth without bitterness, burning sensation in the eye corners, tidal heat, sneezing, migraine, borborygmi, abdominal distension, poor appetite, ulnar pulse (TCM) and pulse string-like taut. Fig.11 displays the visualization of high-order connections. Our visualization process is akin to the thought process of a real doctor during a diagnosis. In this process, there are two key observations:

The high-order information aggregation process in the SCEIKG model closely resembles the diagnostic approach of a real doctor. The visualization in Fig.11 can be interpreted as the doctor's contemplation throughout the four diagnostic methods. The symptoms obtained through observation have the related symptom of poor sleep through higher-order connectivity. Similarly, the relevant symptoms inferred from the high-order connectivity of smelled symptoms encompass both poor sleep and the manifestation of a wind-cold common cold. Furthermore, the relevant symptoms obtained through high-order connectivity of inquired symptoms indicate weaknesses in the spleen and stomach. Lastly, the relevant symptoms acquired through palpation point to an excess of liver yang. Our primary objective is to aggregate information at a higher level. Given the abundance of entities and relations in the knowledge graph, we systematically select and cumulatively integrate the results of two-order connectivity. For instance, migraine → general weakness → wind-cold and flu, we aggregate the scores of these two two-hop connectivity. In Fig.11, the scores of herbs such as Guizhi, Dazao, and Renshen emerge as higher, aligning with the herbs commonly prescribed by real doctors. Due to the multitude of herbs involved, we refrain from displaying the scores of all herbs.

Table 7: Experimental results of without sequences SCEIKG variants.

| Ablation | Precision | | | Recall | | | F1 | | |
|---|---|---|---|---|---|---|---|---|---|
| | P@5 | P@10 | P@20 | R@5 | R@10 | R@20 | F1@5 | F1@10 | F1@20 |
| w/o Sequence1 | 0.2617 | 0.1490 | 0.1215 | 0.1249 | 0.1458 | 0.2252 | 0.1652 | 0.1432 | 0.1543 |
| w/o Sequence2 | 0.4282 | 0.3906 | 0.2872 | 0.2296 | 0.4053 | 0.5683 | 0.2872 | 0.3836 | 0.3708 |

The second key point is the crucial importance of the quality of the knowledge graph. As observed, the scores in our weight matrix are very small. This inspires us to pay closer attention to constructing the knowledge graph in future work, especially in filtering entities with limited information.

**Poor Performance without sequences** The performance in the absence of sequence data is indeed an area we aimed to explore to demonstrate the additional value of historical context in traditional Chinese medicine recommendations. Traditional Chinese medical practices often rely on a comprehensive understanding of a patient's condition over an extended period, including the evolution of symptoms and treatment responses. The primary innovation of our model lies in leveraging this sequential data to enhance accuracy. Through experiments, we discovered that when we did not utilize sequences for herbal recommendations, incorporating the overall patient condition into the model led to the results shown in Fig.3. Subsequently, we removed the overall patient condition and conducted experiments again using a model without sequence information, and the experimental results in Table 7 show that we are effective in recommending without using historical information. w/o Sequence1 is the result of the ablation experiment of SCEIKG without sequence in our paper, and w/o Sequence2 is likewise the result of the ablation experiment of SCEIKG without sequence, but with the overall condition of the patient removed from the experiment. We analyze this phenomenon: (1) Role of the Sequence Module, the sequence module may play a crucial role in handling sequential data and removing it resulted in the model losing its ability to process sequence information. The information provided in the textual description may have been better integrated and utilized in the sequence module. In the absence of sequences, noise in the disease features may lead to interference

Table 8: The difference and intersection herbs prescribed by SCEIKG, w/o IKG and TCM doctor according to clinical symptoms and records of the same patient for two diagnoses.

| Sequential diagnoses | Symptom Set | Herb Set | | |
|---|---|---|---|---|
| | | w/o IKG | SCEIKG | TCM doctor |
| First diagnosis | 抑郁症 (depression)
口干 (xerostomia)
大便费力 (dyschezia)
入睡困难 (insomnia)
眠浅易醒 (light sleep, easy to wake up)
乏力 (fatigue)
胸闷 (chest tightness)
四肢麻木 (numbness of limbs)
舌淡红 (pale red tongue)
下睑淡白 (pale lower eyelid) | 黄芩 (scutellaria baicalensis)
炙甘草 (glycyrrhiza uralensis)
生姜 (ginger)
大枣 (jujube)
人参 (ginsen)
桂枝 (cinnamomum cassia)
茯苓 (tuckahoe)
川芎 (sichuan lovage rhizome)
法半夏 (pinellia tuber)
当归 (angelica sinensis (Oliv.) diels) | 黄芩 (scutellaria baicalensis)
炙甘草 (glycyrrhiza uralensis)
生姜 (ginger)
大枣 (jujube)
人参 (ginsen)
茯苓 (radix adenopharae)
茯苓 (bupleuri radix)
白芍 (paeonia lactiflora)
牡蛎 (ostrea gigas)
干姜 (zingiber officinale) | 黄芩 (scutellaria baicalensis)
炙甘草 (glycyrrhiza uralensis)
生姜 (ginger)
大枣 (jujube)
人参 (ginsen)
北沙参 (radix adenopharae)
柴胡 (bupleuri radix)
天花粉 (flos rosae rugosae) |
| Second diagnosis | 口干 (xerostomia)
惊恐 (panic)
焦虑 (anxiety)
入睡困难 (insomnia)
眠浅易醒 (light sleep, easy to wake up)
乏力 (fatigue)
胸闷 (chest tightness)
四肢麻木 (numbness of limbs)
小便频急 (frequent urination)
右手心热 (palm heat)
舌淡红 (pale red tongue)
苔薄 (thin fur)
下睑淡白边偏红 (pale lower eyelid with reddish edges) | 黄芩 (scutellaria baicalensis)
赤芍 (paeonia lactiflora)
炙甘草 (glycyrrhiza uralensis)
大枣 (jujube)
生姜 (ginger)
茯苓 (tuckahoe)
人参 (ginsen)
桂枝 (cinnamomum cassia)
甘草 (glycyrrhiza uralensis fisch)
当归 (angelica sinensis (Oliv.) diels) | 黄芩 (scutellaria baicalensis)
赤芍 (paeonia lactiflora)
炙甘草 (glycyrrhiza uralensis)
大枣 (jujube)
生姜 (ginger)
清半夏 (pinellia tuber)
人参 (ginsen)
桂枝 (cinnamomum cassia)
茯苓 (bupleuri radix)
炒六神曲 (medicated leaven) | 黄芩 (scutellaria baicalensis)
赤芍 (paeonia lactiflora)
炙甘草 (glycyrrhiza uralensis)
大枣 (jujube)
生姜 (ginger)
清半夏 (pinellia tuber) |

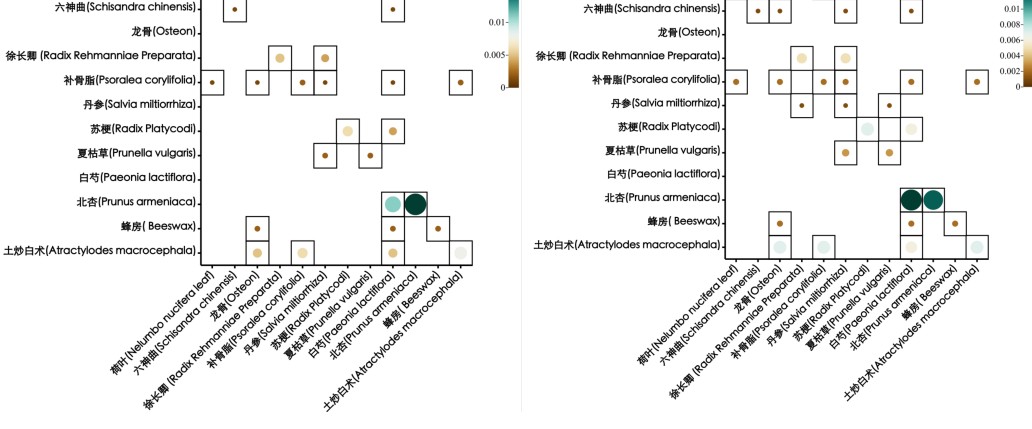

Figure 12: The visualization of the heatmaps on the relationship between partly herbal pairs. (a) Partly herb pairs, derived from a constructed weight matrix that captures the co-occurrence of all entities involved; (b) Partly herb pairs, through the training of our SCEIKG model.

from redundant information, thereby reducing performance; (2) Association between Features and Sequences, the inclusion of overall patient condition as a feature in the model may depend on certain patterns or contextual information in the sequence data. Removing the sequence module may hinder the model's ability to correctly capture these associations, resulting in a decline in performance.

## E  ADDITIONAL CASE STUDY

**Herb Recommendation** We observed that SCEIKG slightly underperforms SCEIKG without IKG on certain metrics in Fig.3. In Table 8, we highlighted herbal recommendations consistent with real TCM doctors in red and indicated inconsistency between the recommendations of the two models in blue. However, However, the evaluation by real doctors indicates that the recommendations generated by SCEIKG with IKG align more closely with classical prescriptions found in ancient texts, which have been validated over thousands of years and are considered more reliable. This suggests that the recommendations from the SCEIKG model with IKG are more reasonable and align better with traditional knowledge. Additionally, IKG provides richer information for TCM recommendations, while the model without IKG relies solely on basic data for recommendations. This also underscores the rationality of herbal compatibility. From the perspective of herbal combinations, IKG furnishes more in-depth and comprehensive information for TCM recommendations, facilitating better decision-making by medical professionals.

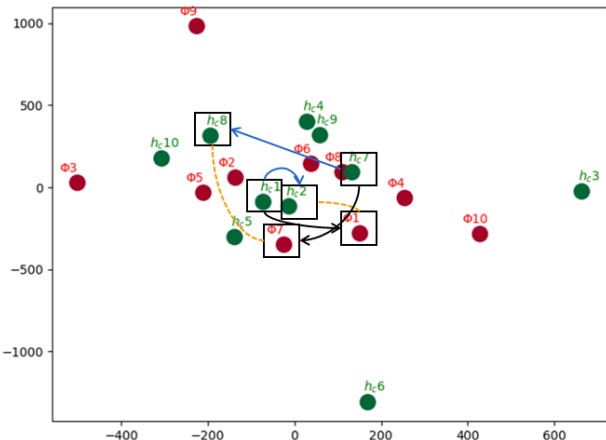

Figure 13: Conditional embedding visualization. The black solid line represents the transformation of the patient's current diagnosis status to the status after taking medication, and the blue solid line indicates the transition from the current diagnosis status to the status at the next diagnosis. On the other hand, the orange dashed line represents the extent to which the current status after taking medication transfers to the status at the next diagnosis.

**Herb Compatibility** As illustrated in Fig.12, we unveil the shift in the correlation between partly herb pairs, which captures the co-occurrence of all entities involved. For instance, the connection between (schisandra chinensis, osteon) and (radix rehmanniae preparata, salvia miltiorrhiza). These once-disparate pairs now harmoniously coexist within the same prescription. The change reverberates as a testament to the constraints imposed on the compatibility of these herb pairs. The interpretative experiments of the embedding visualization and herb recommendations on the ZzzTCM dataset are given in the supplementary material.

**Embedding Visualization** To show a more intuitive understanding of the changes in patient condition. We utilized the t-SNE(Van der Maaten & Hinton, 2008) to portray the patient's real condition embeddings $h_c$ and horizontal patient condition embeddings $\Phi$. As shown in Fig.13, it becomes apparent that patients' condition changes tend to cluster together, while also allowing for some isolated instances. This intriguing phenomenon arises from the limited correlation observed between a patient's current diagnosis and their previous diagnoses. The patient's condition $h_c^1$ during their initial diagnosis. As the patient follows the prescribed medication, a state transition embedding, referred to as $\Phi^1$, occurs, and we observed a relatively small degree of state transfer from $h_c^1$ to both $\Phi^1$ and $h_c^2$. In subsequent diagnoses, we can observe a significant distance discernible between the state $\Phi^7$ after the transfer of $h_c^7$ to the predicted state and the state $h_c^8$ at the next diagnosis, with different degrees of state transfer used to assist the current true state for medication recommendation, thus improving the accuracy of the recommendation. The difference in the degree of state transfer is due to the fact that the patient will not respond to the recommended remedy to the same degree. However, we use implicit state transfer to assist in subsequent diagnoses, and in future work, we will represent our states in a more direct way.

