# OpenReview forum: "Sequential Condition Evolved Interaction Knowledge Graph for Traditional Chinese Medicine Recommendation"
_ICLR.cc/2024/Conference — Submitted to ICLR 2024_

### Official Review · Reviewer_K5T8 · 2023-10-21

**Soundness:** 3 good
**Presentation:** 3 good
**Contribution:** 3 good
**Rating:** 6
**Confidence:** 4

**Summary:**

The existing Chinese medicine recommendation methods ignore the changes in the patient's condition. The author of this article proposes a Sequential Condition Evolved Interaction Knowledge Graph(SCEIKG), which comprehensively considers the dynamic changes in the patient's condition and formulates questions in a sequential manner. It also takes into account the effects of different herbs and the patient's condition to enhance the accuracy of the recommendations.

**Strengths:**

It comprehensively considers the dynamic changes in the patient's condition and innovatively proposes a framework to address this issue.
By adding an interactive knowledge graph to improve accuracy is an interesting contribution. Experimental results demonstrate that this method outperforms current approaches.

**Weaknesses:**

In Section 5， the dataset used in this study is relatively small, covering only 751 patient records and prescriptions, all of which are from Guangdong province. This may result in limitations to the model's generalizability. The patient’s state transfer is implicit so it may be professional errors in the dataset that are difficult to detect. The paper can discuss the limitations and potential biases of the dataset.

In Appendix A.1， Insufficient evaluation of the knowledge graph: This article utilizes a knowledge graph embedding method that combines TransR with RotatE.Different knowledge graph completion methods may have different impacts on the results. This article does not provide comprehensive evaluation results for the generated knowledge graph. The quality and accuracy of the knowledge graph are very important for the subsequent tasks.

In Section 4， this article introduces a module to predict changes in patient condition but does not explain the accuracy of this prediction. It can also clarify how SCEIKG handles patient-specific changes to better understand and model individual differences, which may help improve the accuracy of the model and personalized recommendation ability.

In Appendix D， This article mentions the issue of herb compatibility in herbal recommendations but does not provide specific solutions to address this compatibility problem.

Overall, the paper presents an innovative approach to TCM recommendation, but further clarity and validation are needed. The use of interaction graphs is promising, but practical implications should be explored in more detail.

**Questions:**

Can we increase the reason for recommendations to enhance the interpretability of the model? The interpretability of the model is very important for both doctors and patients, which may enhance its practicality.

---

> ### Author Response · Authors · 2023-11-18
>
> **1.Dataset Size and Generalizability**
>
> Regarding the reviewer's concern about the small size of the experimental dataset, we would like to address the issue from the following three aspects.
>
> Firstly, we acknowledge that the dataset utilized in this paper is relatively small. Due to the subdivision of traditional Chinese medicine (TCM) and concerns related to patient privacy, the scale of TCM datasets is indeed constrained. **However, the TCM sequence dataset we utilized is extremely valuable, accumulated over a decade from the Guangdong Provincial Hospital of Chinese Medicine, one of China's top hospitals. It encompasses patient information from various regions of China, not limited to Guangdong Province.**  Additionally, publicly available datasets for TCM recommendations, such as TCM, typically only include information on symptoms and herbs, making them relatively limited for research on TCM sequence recommendations. Therefore, despite the small dataset size, these data hold significant value in the context of TCM research and provide ample diversity for validating the SCEIKG model. Furthermore, we plan to gradually expand the dataset size while ensuring patient privacy and data quality as more data becomes available.
>
> Secondly, despite the small size of our dataset, during model training, we not only utilized the ZzzTCM dataset but also trained on the IKG. The knowledge graph we constructed contains 4,308,799 triplets, involving 344,092 entities and 35 types of edge relationships. Both the TCM dataset and the ZzzTCM dataset were incorporated into the construction of the IKG. Detailed descriptions of the IKG are provided in Appendix A.1, and Appendix Table 4 outlines the data information, indicating the relative richness of our data resources.
>
> Finally, we conducted additional experiments where we partitioned the ZzzTCM training data at different ratios to assess the robustness of our model in situations with a relatively small dataset. The results are presented in the table below,
>
> | TrainingDataset | Precision@5 | Precision@10 | Precision@20 | Recall@5 | Recall@10 | Recall@20 | F1@5 | F1@10 | F1@20 |
> |------------------|-------------|--------------|---------------|----------|-----------|-----------|------|-------|-------|
> | 0.2Train         | 0.4813      | 0.3919       | 0.2809        | 0.2454   | 0.3913    | 0.5514    | 0.3164 | 0.3803 | 0.3627 |
> | 0.4Train         | 0.4648      | 0.3847       | 0.2912        | 0.2352   | 0.3864    | 0.5722    | 0.3041 | 0.3741 | 0.3758 |
> | 0.6Train         | 0.4933      | 0.3960       | 0.2951        | 0.2571   | 0.4088    | 0.5933    | 0.3287 | 0.3901 | 0.3835 |
> | 0.8Train         | 0.4959      | 0.4025       | 0.2879        | 0.2551   | 0.4089    | 0.5733    | 0.3264 | 0.3931 | 0.3721 |
> | 0.9Train         | 0.5084      | 0.4206       | 0.3040        | 0.2470   | 0.4021    | 0.5718    | 0.3235 | 0.3994 | 0.3867 |
> | 0.94Train        | 0.5477      | 0.4275       | 0.3087        | 0.2727   | 0.4243    | 0.6010    | 0.3538 | 0.4128 | 0.3973 |
> | 0.98Train        | 0.6000      | 0.4579       | 0.3066        | 0.3059   | 0.4613    | 0.6151    | 0.3932 | 0.4465 | 0.3987 |
>
> From these results, it is evident that with an increase in the training set, the evaluation metrics show an upward trend to some extent. This indicates that the dataset size is not the sole determining factor for the model's performance. **We added this experiment to the Appendix C Parameter Sensitivity section of the revised version, at line 607 of the paper, and drew the trend of the results, displayed in Figures 6, 7, and 8 of the paper, and highlighted in red. See the revised version for details.**
>
> **With regard to the potential for professional errors in the dataset, we processed the data by firstly using ChatGPT's advanced language comprehension capabilities to parse and extract the symptom set from the historical records, and then we utilized the expertise of experienced Chinese medicine practitioners to sift through our data, an approach that reduces the risk of professional errors in the dataset. However, we agree that this risk is an important consideration.**  In addition, the dataset was primarily collected for the primary treatment of insomnia and, as we can see from Fig.5 in our Appendix A, the top ten symptoms and herbs in our dataset were all related to insomnia, and different types of illnesses were not considered in our dataset. We discuss the limitations of our dataset in the revised version of Conclusion.
>
> Thank you again for your review and valuable comments and we will continue to work on improving and refining our study.

---

> ### Author Response · Authors · 2023-11-18
>
> **2. Knowledge Graph Evaluation**
>
> We appreciate the careful review by the Reviewer regarding the knowledge graph evaluation in our study.  We recognize that a more thorough evaluation of this component can strengthen our research findings and provide a clearer understanding of the impact of the knowledge graph on our results. To address the issues raised by the reviewer, we have supplemented with additional Knowledge Graph Completion (KGC) methods. This allows us to compare and contrast the effectiveness of the methods we adopted with other approaches, providing a more comprehensive understanding of the influence of the IKG on model performance. **We conducted a supplementary Evaluation of IKG in Revision 4.2, at line 302 of the paper, where we experimented with various knowledge graph completion methods, and further confirmed the validity of our method by the results highlighted in orange in Table 3.** The results are as follows,
>
> | Methods     | Precision@5 | Precision@10 | Precision@20 | Recall@5 | Recall@10 | Recall@20 | F1@5 | F1@10 | F1@20 |
> |-------------|-------------|--------------|---------------|----------|-----------|-----------|------|-------|-------|
> | TransR       | 0.4738      | 0.3658       | 0.2695        | 0.2517   | 0.3818    | 0.5475    | 0.3159 | 0.3602 | 0.3508 |
> | TransE       | 0.5329      | 0.4315       | 0.3074        | 0.2680   | 0.4360    | 0.6077    | 0.3470 | 0.4211 | 0.3964 |
> | ComplEx      | 0.4966      | 0.4128       | 0.2936        | 0.2531   | 0.4258    | 0.5844    | 0.3260 | 0.4044 | 0.3798 |
> | RotatE       | 0.5114      | 0.3960       | 0.2889        | 0.2562   | 0.3903    | 0.5596    | 0.3316 | 0.3811 | 0.3713 |
> | $\rho$RotatE | 0.5396      | 0.4396       | 0.3111        | 0.2687   | 0.4382    | 0.6131    | 0.3491 | 0.4261 | 0.4011 |
> | DistMult     | 0.4153      | 0.3980       | 0.2893        | 0.2490   | 0.4002    | 0.5673    | 0.3215 | 0.3866 | 0.3731 |
> | Our          | 0.5477      | 0.4275       | 0.3087        | 0.2727   | 0.4243    | 0.6010    | 0.3538 | 0.4128 | 0.3973 |
>
> From the results, it can be observed that different knowledge graph complementation methods have an impact on our herbal recommendation results, and it can also be seen that the TransR combined with RotatE method that we have used has the best results, which further proves the effectiveness of our method.
>
> We thank the reviewers for their constructive feedback.

---

> ### Author Response · Authors · 2023-11-18
>
> **3. Herb Compatibility**
>
> **Thanking the reviewers for their questions, we have considered the issue of herb compatibility during model iteration, while we use domain knowledge for constraints to improve herb suitability and safety. While SCEIKG does not currently explicitly model DDI, it implicitly addresses the issue of herb compatibility through an iterative interaction knowledge graph that includes relationships between different herbs.** We have added a new experimental analysis of Herb Compatibility to section 4.2 of the revised version, which is demonstrated by Figure 4, in line 263 of the paper, and highlighted in red. In addition, we have reflected on the results of herb interactions by analyzing real doctors. We will provide you with further analyses.
>
> In the motivation section of our paper, we highlight a significant and perplexing issue in TCM, which is the compatibility of herbal medicines, specifically, the interactions between different herbs when combined. However, the TCM treatment process encompasses a vast amount of complex TCM knowledge and involves four steps, making it challenging to elucidate the intricate interactions between different herbs. **To address this issue, we propose an approach based on domain knowledge from a knowledge graph to ensure herbal compatibility. Nevertheless, it's important to note that due to the complexity of TCM and the lack of standardization, accurately determining interactions between herbs can be challenging. For instance, herbs A and B may have antagonistic effects, but they might reach equilibrium when combined with herb C.** These interaction criteria are highly intricate, and herb compatibility is not fixed. Therefore, to enhance the accuracy of herbal recommendations, we constrain TCM recommendations based on domain knowledge from traditional Chinese medicine. Specifically, we introduce a regularization scheme
>    $\boldsymbol{R}$
>    $=-\sum_{i=1}^{N}\sum_{j=1}^{N}W_{ij}^{h}\hat{hc}^{(t)}_i \hat{hc}^{(t)}_j$ to penalize $P^{(t)}$, limiting violations of herb pairs. The $W^{I}$ derived $W{i,j}^{h}$
>    weights from $\mathcal{G}^{I}$ represent the strength of compatibility between the i-th and j-th herbs. If they are mutually exclusive, $W{i,j}^{h}=0$.
>
> We demonstrate that we reduce the risk of herbal incompatibility by both，
>
> First, in the Herb Compatibility section of Appendix E of this paper, on line 679 of the paper, we conduct a detailed result analysis. Due to data privacy and confidentiality, we present partial correlations between herb pairs in the appendix, which dynamically change during model training. **According to real doctors' analysis, these changes are reasonable as they adapt to the patient's current condition. In Table 8, we showcase the results of herbal recommendations.  Although there are differences between the herbs we recommend during the first diagnosis and those prescribed by doctors, real doctors' analysis indicates that the model-recommended prescription is better suited to the patient's current condition. During the second diagnosis, although there are differences between the herbal recommendations and the actual prescription, doctors point out that the gap between them is not significant, and there are no mutually antagonistic herbs.** This further validates the effectiveness of considering herb compatibility and emphasizes that herb compatibility is a complex and dynamic issue.
>
> Secondly, **the experimental analyses were added to 4.2 in our revised version, in line 263 of the paper, highlighted in red.** Fig. 4 illustrates the heatmap for all herb pairs; herb names are omitted due to data privacy. From the heatmap, we observe that the magnified positions, specifically (Cassia Bark(肉桂)), Red Halloysite(赤石脂)), and (Danshen Root(丹参), Lightyellow Sophora Root(苦参)), both exhibit a correlation of 0. The correlation between "Cassia Bark(肉桂)" and "Red Halloysite(赤石脂)" is 0, indicating a certain degree of mutual antagonism between these two herbs. This aligns with the ancient literature's viewpoint that "Cassia Bark is effective in regulating cold energy, but it loses its efficacy when encountered with Red Halloysite(官桂善能调冷气，若逢石脂便相欺)." In other words, Cinnamon and Red Halloysite are mutually repellent. Additionally, "Danshen Root(丹参)" and "Lightyellow Sophora Root(苦参)" cannot be mixed in some situations due to their differing medicinal properties. Nevertheless, real doctors also point out that some herb combinations may be controversial as they may have different effects in clinical practice. **We acknowledge that we have not yet fully addressed the issue of herb interactions**, but we have selected **70** recommended results for real doctors to analyze, and after real doctors' analysis, we have achieved a **91.4%** level of compatibility for our recommended herb pairings, which, in terms of our recommended results, demonstrates the validity of our method.

---

> > ### Author Response · Authors · 2023-11-21
> >
> > **4. Recommendation Interpretability**
> >
> > To further explain our recommendation results and the interpretability of the model, we employ higher-order connectivity reasoning to infer herbs for a patient's current condition. It is worth noting that given that both symptoms and remedies are in set.  The higher-order connectivity is based on the IKG's weight matrix values. We select entities with larger neighboring weights for information aggregation rather than simply showing path selection.
> >
> > We randomly selected a patient's record from the ZzzTCM dataset, and for privacy reasons, we briefly describe the patient's symptoms: **{lower lids{pale redness in the lower eyelid, white and greasy coating on the tongue, dark red tongue, dry mouth without bitterness, burning sensation in the eye corners, tidal heat, sneezing, migraine, borborygmi, abdominal distension, poor appetite, ulnar pulse (TCM) and pulse string-like taut.}**  Fig.11 in this paper shows the visualization of higher-order connections. Our visualization process is similar to the thought process of a real doctor when making a diagnosis and can be understood as the entire thinking process of a doctor in the four diagnostic methods. The symptoms obtained through observation have the related symptom of poor sleep through higher-order connectivity. Similarly, the relevant symptoms inferred from the high-order connectivity of smelled symptoms encompass both poor sleep and the manifestation of a wind-cold common cold. Furthermore, the relevant symptoms obtained through high-order connectivity of inquired symptoms indicate weaknesses in the spleen and stomach. Lastly, the relevant symptoms acquired through palpation point to an excess of liver yang.  Our main goal is to aggregate higher-order information. Given the large number of entities and relationships in the knowledge graph, we systematically selected and cumulatively integrated the results of two-hop connectivity. For example, for migraine$\rightarrow$ generalized weakness$\rightarrow$ wind-cold flu, we aggregated the scores for both two-hop connections. In Fig.11, herbs such as cinnamon sticks, jujube and ginseng scored higher, which is consistent with the common prescriptions of real doctors. Due to the large number of herbs involved, we do not show scores for all herbs.
> >
> > **We added the recommended interpretability experiment in Appendix D of the revised version, at line 615 of the paper, highlighted it in orange, and gave Fig.11 for demonstration. We use a patient and information fusion by the patient's symptoms to find multi-hop connectivity based on the IKG as a way to justify the edges of our IKG. See line 615 of the revised version for a detailed analysis.**
> >
> > We thank the reviewers for their constructive feedback.

---

> > > ### Author Response · Authors · 2023-11-23
> > > **Questions remaining?**
> > >
> > > Dear R K5T8
> > >
> > > Does our response address your concerns? Do you have any remaining questions or concerns following our response? Please let us know. We’d be very happy to do anything we can that would be helpful in the time remaining! We hope the score could be raised if there is no further concern about our work. Thank you very much.
> > >
> > > The authors.

---

### Official Review · Reviewer_WEMc · 2023-10-25

**Soundness:** 2 fair
**Presentation:** 2 fair
**Contribution:** 2 fair
**Rating:** 5
**Confidence:** 3

**Summary:**

The paper focus on the problem of Traditional Chinese Medicine Recommendation which is formalized as a sequential prescription-making problem. Further, the paper introduces a methods named SCEIKG, which considers dynamics of patients' conditions across multiple diagnoses. In addition, an interaction knowledge graph is utilized to enhance the accuracy of recommendations. The SCEIKG contains three major modules, a heterogeneous graph neural network which combines a hierarchical attention message passing layer and a knowledge graph embedding layer, a horizontal conditional module which is used to extract the patient’s status, and a transition condition module to model the patients’ progression. The experiments are conducted on one real-world dataset to show the proposed methods' performance.

**Strengths:**

1.The paper focus on the problem of Tradition Chinese Medicine Recommendation, which is an important problem in medical data mining literature.
2.The proposed method utilizes domain knowledge to enhance the recommendation of traditional Chinese medicine.
3.The experimental results show the advantages of the proposed method on the real-world dataset.

**Weaknesses:**

1.The motivations of the paper needs to be clarified. The differences between Traditional Chinese Medicine Recommendation and General Medicine Recommendation should be emphasized. The author may want to explain these differences. Besides, it seems that the proposed method can be adopted into any other medicine recommendation setting. The author may need to stress the motivations.
2.The novelty of the paper is limited. The progress of knowledge fusion is by introducing a regularization into the loss function, which is trivial in the literature. Besides, what is the motivation of the module TRANSITION CONDITION MODULE?  Actually, modeling patients’ dynamics in drug recommendation has been already addressed, such as. The author may need to discuss the difference.
3.The structure of the paper needs to be improved. The method is called Sequential Condition Evolved Interaction Knowledge Graph Learning. However, there is no formal definition of what an Interaction Knowledge Graph(IKG) is. The author may want to declare it.
4.For experiments, the method is only conducted on one dataset. Evaluations on more dataset should be conducted. The proposed method seems to be a general framework for drug recommendation. Therefore, the author may need to show the results on common datasets.

**Questions:**

1. What is IKG？ Is there any difference between common KGs and IKGs?
2. What is the difference between Traditional Medicine Recommendation and general medicine recommendation? It seems that the proposed method can be used in general medicine recommendation setting. The author may need to point out the differences.

---

> ### Author Response · Authors · 2023-11-17
>
> 1. **Motivation of the paper**
>
> The main motivations behind our research are as shown below **(which were also addressed in the second paragraph of the introduction section):**
>
>  **Lack of Sequence Data Research in Traditional Chinese Medicine (TCM):** We observed a scarcity of studies focusing on sequential data analysis in the TCM, especially when it comes to incorporating domain knowledge into herbal recommendations. Most existing TCM recommendation studies heavily rely on TCM public datasets, which are not sequential data. Furthermore, many TCM-based studies merely utilize the dataset itself without dynamically integrating domain knowledge.
>
> **Uniqueness and Specificity of Traditional Chinese Medicine (TCM):** Traditional Chinese medicine differs significantly from general Western medicine in terms of treatment methods and diagnostic processes (often presented by unstructured data, e.g., text). TCM emphasizes personalized and holistic approaches, often requiring intricate herbal combinations tailored to an individual's overall condition. In contrast, Western medicine leans more towards quantifiable metrics, such as laboratory test results. TCM relies on comprehensive patient evaluations through observation, olfaction, inquiry, and pulse diagnosis, which distinguishes it from general medicine. **While some recommendation methods may be applicable in other medical domains, our work is tailored specifically to the field of TCM, with domain knowledge that leans towards traditional Chinese medicine principles.**
>
>    **In summary, our research is motivated by two key factors:**
>
>    **Firstly, we have developed a model to effectively capture the state transitions of patients following herb treatments.** In traditional sequence data analysis, the focus primarily revolves around changes in a patient's condition between different diagnoses, often involving comparisons of metrics and data collected at these diagnoses. However, this approach may not adequately account for the effects of herb treatments, as it mainly relies on comparisons of diagnosis information. In contrast, our model provides a more comprehensive method for modeling state transitions following herb treatments. We consider changes in a patient's status over a time series, allowing us to observe the evolution of a patient's condition over an extended period, including the impact of herb treatments. This approach enhances our understanding of the efficacy of TCM by considering state transitions, ultimately improving the accuracy of herbal recommendations.
>
>    **Secondly, we incorporate domain-specific knowledge from the traditional Chinese medicine into our model training process.** This knowledge integration is not limited to pre-trained information but involves a dynamic utilization of domain expertise. This fusion of knowledge provides our approach with a significant advantage in the field of traditional Chinese medicine.
>
>    Through these innovative methods, our aim is to provide more rational herbal recommendations within the domain of traditional Chinese medicine. **We will elaborate further on the innovation and advantages of our paper in the following respones.**

---

> ### Author Response · Authors · 2023-11-17
>
> 2. **Novelty of the Paper:**
>
> We sincerely appreciate the reviewer's evaluation of our work. We would like to further elaborate on the innovations in our research. **The novelty of our work lies in the introduction of the Interaction Knowledge Graph (IKG), which integrates domain knowledge from TCM through an interactive knowledge graph and applies it to sequential herbal medicine recommendations.This approach, combined with our transition condition module, which explicitly models the evolution of patients' conditions, sets our work apart from existing drug recommendation systems that may not capture these dynamic aspects​​.**
>
>    **Firstly: Incorporation of Domain Knowledge and Knowledge Graph Completion** We fully recognize that constructing a comprehensive knowledge graph for TCM is a great challenge that requires substantial data support. In our study, we employed knowledge graph completion to infer new facts from known facts, enriching the knowledge graph and providing more information for herbal medicine recommendations. Training the knowledge graph embeddings together with the network is distinct from traditional recommendation systems, which typically focus on modeling individual users and items. In our case, we need to consider symptom sets on one side and herb sets on the other, which vary in size and pose a unique challenge. This comprehensive approach allows us to more accurately represent and utilize domain-specific knowledge in the field of TCM. **Additionally, we conducted a supplementary Evaluation of IKG in Revision 4.2, at line 302 of the paper, where we experimented with various knowledge graph completion methods, and further confirmed the validity of our method by the results highlighted in orange in Table 3.**
>
>    **Secondly: Modeling Based on Sequential Data and State Transitions** We deeply understand the significance of modeling patient state transitions after herbs in the TCM. In our research, we introduced latent state representations to capture changes in patient conditions after taking prescriptions. This process is crucial in real-life scenarios, as doctors typically recommend herbs based on a patient's past diagnosis history and current condition. Our approach involves predicting patient states post-herbs and leveraging the potential inference inthe state transition process. Even in cases where we lack comprehensive records of a patient's next diagnosis, our method allows us to predict a patient's conditional representation, enabling us to make reasonable herb recommendations. **This aspect, which has been less considered in prior research on sequence data-based medication recommendations, is pivotal for better recommendations**.
>
>    The update process of SCEIKG resembles the concept of the EM algorithm.The updates to IKG will iteratively contribute to improving various modules depicted in Fig.2, and the model training within Fig.2 will reciprocally enhance IKG (on the right side of Fig.2). These two components collaborate and drive the entire iterative updating process.The incorporation of domain-specific knowledge into the loss function inthe TCM domain is not merely adding a regularization term but involves theutilization of knowledge graph completion during the update of IKG.
>
>    In conclusion, our research comprehensively considers the fusion of domain knowledge and sequential data to enhance the accuracy of herbal recommendations. We would like to express our gratitude once again to thereviewer for their valuable suggestions and questions, which have stimulated deeper thinking and provided directions for further research.

---

> ### Author Response · Authors · 2023-11-17
>
> 3. **Definition of IKG and  Differences between IKG and Common KGs:**
>
> **Definition of IKG:** We define IKG in the revised version of Section 2, Problem Formulation, on line 93 of the paper and further elaborated in Appendix A.1. In this paper, for clarity, we highlight in blue.  **We define an IKG by $\mathcal{G}=(\mathcal{E},\mathcal{R},\mathcal{T}, \mathcal{A})$, where $\mathcal{E}$ is a set of entities and $\mathcal{R}$ is a set of relations. $\mathcal{T}$ is a set of triples $\mathcal{T}=\{(h,r,t)|h \in \mathcal{E}, t\in \mathcal{E}, r\in \mathcal{R}\}$, where each triples means there is a relation $r$ from *head* entity $h$ to *tail* entity $t$. Specifically, $\mathcal{E}$ consists of symptoms $\boldsymbol{S}$, herbs $\boldsymbol{H}$, and other entities such as pharmacology, efficacy, diseases, examination and diagnosis, which were extracted from TCM datasets to help entail relations between symptoms and herbs directly or indirectly.**
>
>  **The differences between KG (Knowledge Graph) and IKG (Interaction Knowledge Graph) are as follows:**
>
> **Data Source:** KG is initially constructed based on data from multiple sources, while IKG is an enhanced version built on top of it. KG may not include all the symptoms and herbal entities from the dataset, as it is constructed from multiple sources and may have missing elements. To ensure the quality and completeness of the knowledge graph, additional measures are taken for KG. These measures include adding symptoms and herbal entities present in the dataset and constructing corresponding triplets to fill potential gaps.
>    **Triplets:** In the construction of IKG, we employed the Normalized Pointwise Mutual Information to enhance the knowledge graph, improving its connectivity.
>
> In summary, IKG is an enhanced and optimized knowledge graph built upon KG to improve its quality, completeness, and semantic relationships. It includes more entities and relationships to better support applications and research in the field of Traditional Chinese Medicine.

---

> ### Author Response · Authors · 2023-11-17
>
> 4. **The motivation of the module Transition Condition Module**
>
> We deeply appreciate the reviewer's insightful comments and concerns regarding our Transition Condition Module. To address these points comprehensively, we provide an expanded explanation of this module in conjunction with the Horizontal Patient Representation, elaborating its mechanism and motivation. In Eq.(4), $\Phi^{(t)}_P = \boldsymbol{LSTM} \left( \left(  \Pi \left(h_S^{(t)},\Psi^{(t-1)}\right) ,C^{(t-1)} \right),...,\left( \Pi \left(h_S^{(0)},h_C^{(0)}\right) ,C^{(0)}\right); \; W_6\right)$
>
> We employ LSTM for sequential prediction, where $\Phi^{(t)}$ denotes the comprehensive patient state during the $t$-th diagnosis. This state includes the patient's historical condition representation $h_C$, symptoms representation $h_S$, and the transitional state $\Phi^{(t-1)}$, derived dynamically through the state transition function
>
>  $ \boldsymbol{T} \left(\Phi^{(t)}_P,\boldsymbol{P}\left(\Phi^{(t)}_P, E_h\right)\right)$
>
> $= \boldsymbol{NN_4} \left(Concatenate\left({\Phi_{P}^{(t)},(\Phi_{P}^{(t)}\odot  h_{H}^{(t)}),h_{H}^{(t)}}\right);\; W_8\right)$.
>
> **To offer a more intuitive illustration,** consider a scenario involving three diagnoses. During the first diagnosis, leveraging the patient's current condition representation $h_C^{(1)}$ and symptoms representation $h_S^{(1)}$, we compute the comprehensive patient state $Φ^{(1)}$, upon which the initial herbal recommendation is based. In subsequent diagnoses, the previous hidden state $C^{(1)}$ (retaining the first visit's state information) and the transitional state $Ψ^{(1)}=T(Φ^{(1)}, hc^{(1)})$  are integrated through the transition function $T$, alongside the symptoms representation $h_S^{(2)}$ of the second diagnosis, to derive the second comprehensive patient state $Φ^{(2)}$. Importantly, the second comprehensive patient state $Φ^{(2)}$ contains information from the first diagnosis, facilitating dynamic herbal recommendations. This incorporation enriches the patient's inherent state information, amalgamating the current and preceding diagnosis states, along with the transitional state after herbs. Consequently, this augmentation contributes to enhanced model performance, facilitating improved herbal recommendations.
>
>    Also, the concept of the patient state encapsulates a holistic representation of the patient's well-being, thereby affording a deeper understanding of their condition. **For instance,** a patient's description of "recent severe insomnia accompanied by mild headaches" may not be fully comprehended based solely on symptom representation. Our model predicts the transitional state to address the dynamic nature of herbal effects, considering patient-specific variations and herb interactions. This underpins our rationale for incorporating the transitional state even in cases where the patient's subsequent diagnosis patient state is unavailable, reaffirming our model's robustness in herbal recommendation.
>
> ****In the ablation experiments section of Section 4.2 of the revised version, in line 309 of the paper**, we conducted ablation experiments pertaining to this module.** Specifically, two variants were designed and evaluated: (1) SCEIKG w/o Sequence, which excludes consideration of multiple visits for sequential herb recommendation and the transition condition module, and (2) SCEIKG w/o State, which omits patient condition constraints denoted as $L_{state}$.  As depicted in Fig.3, the performance of both variants lags behind that of SCEIKG, further validating the substantial contribution of this module.
>
> Through the experimental analysis, we can conclude that this Transition condition module plays a very important role in the recommendation of traditional Chinese medicine, which can better consider the patient's condition change and treatment process, thus improving the accuracy of the recommendation of traditional Chinese medicine.

---

> > ### Author Response · Authors · 2023-11-17
> >
> > 5. **Distinction Between TCM and General Medicine Recommendation and Experiments on a Single Dataset**
> >
> > **We appreciate the reviewer's feedback, and we have incorporated their suggestions into the revised version in section of Conclusion, highlighted in blue.**
> >
> >    Traditional Chinese medicine (TCM) and general medicine exhibit differences in drug recommendation, primarily stemming from two aspects: **data sources and treatment philosophies.** Traditional TCM practitioners typically diagnose diseases using methods such as observation, listening, questioning, and pulse-taking, emphasizing the patient's overall condition and symptoms. They rely on the patient's **subjective feelings and the experience of the physician for diagnosis and treatment**. In contrast, general medicine often relies on **modern medical instruments and clinical examinations to diagnose diseases, leaning towards the collection of objective data.** This results in differences in data, with TCM data being more subjective.
> >
> >    **Another significant difference lies in the nature of the drugs.** Western medicine typically has well-defined molecular structures and chemical compositions, whereas herbal medicines consist of parts of natural plants, making their compositions more complex and less easily describable using molecular structures. This complexity necessitates the consideration of different factors in herbal medicine recommendation, such as the efficacy of Chinese herbs, the source of medicinal herbs, and their processing methods. Therefore, there is a fundamental difference between TCM and Western medicine in drug recommendation.
> >
> >    Due to the unique nature of TCM, it significantly differs from general medicine in terms of diagnostic and treatment methods. While our method may have potential applications in other forms of medicine, its design is particularly well-suited for TCM recommendations, given its focus on domain-specific knowledge rooted in TCM principles.
> >
> > **Single Dataset Issue:**
> >
> >    When applying this method to different types of task datasets, such as Western medicine drug recommendations, it is essential to consider the unique characteristics and domain knowledge associated with each dataset. The field of healthcare is extensive and complex, encompassing various disciplines and specialized knowledge. Therefore, significant differences exist between various tasks, including data requirements, evaluation criteria, and real-world application contexts.
> >
> >    It is crucial to emphasize that there are significant differences in medical philosophies and treatment approaches between Traditional Chinese Medicine (TCM) and Western medicine. These distinctions introduce certain limitations in the domain of drug recommendations. Previous studies, such as SafeDrug and GAMNet, have conducted drug recommendation research based on Western medicine datasets. However, the molecular chemical structure information used in these studies is not applicable to herbal medicine recommendations. This underscores the limitations in the scope of applicability of these methods.
> >
> >    When attempting to extend this method to other medical datasets, careful consideration of the differences in data characteristics is essential to ensure that the recommendations align with clinical needs effectively. For publicly available datasets like TCM, which do not necessarily represent sequential data, they may not align with the specific objectives of our research. **However, in future studies, we can explore approaches such as data preprocessing and transfer learning to address these challenges. By flexibly selecting and extracting features, leveraging information from various types of data, and incorporating knowledge accumulated by pre-trained models in relevant domains, we can transfer this knowledge to target tasks. This not only helps broaden the applicability of the method across different domains but also represents a significant direction for our future research.**
> >
> >    We recognize that while our method shows potential in the TCM, its extension to more diverse and wide-ranging datasets requires further in-depth research and experimentation. Exploring the adaptability of the model remains a challenging task for our research and merits deeper consideration in the future.
> >
> >    We would like to express our gratitude once again to the reviewer for their questions, which provide directions for further thinking and exploration.

---

> > > ### Author Response · Authors · 2023-11-23
> > > **Questions remaining?**
> > >
> > > Dear R WEMc
> > >
> > > Does our response address your concerns? Do you have any remaining questions or concerns following our response? Please let us know. We’d be very happy to do anything we can that would be helpful in the time remaining! We hope the score could be raised if there is no further concern about our work. Thank you very much.
> > >
> > > The authors

---

### Official Review · Reviewer_9xVT · 2023-10-30

**Soundness:** 3 good
**Presentation:** 2 fair
**Contribution:** 2 fair
**Rating:** 5
**Confidence:** 3

**Summary:**

This paper claims the traditional Chinese medicine practices are not treating the individual symptoms as well as lack of standardization. The paper points out a. the existing herb recommendation approaches are neglecting the long term effect that could potentially affected by the recommended medication; b. existing approach only relying on diagnose data set and pretrained domain knowledge thus cannot capture the intricate relation between symptom and herb. To address these issues, this paper proposed a three-module herb recommendation system that fuses information form domain knowledge graph, user's diagnosis, symptom, mediation history sequential information, as well as the predicted repercussion of the herb that about to be recommended. The paper reports the empirical comparison between the proposed approach and non-trivial baselines, and achieves best performance in 2/3 of the reported metrics. Case study and module wise ablation study are also provided.

__Initial Recommendation__: Weak reject. Please see weak points for the key reasons.

**Strengths:**

* The proposed three modules are sensible in terms of addressing the claimed existing shortcomings
* The reported experiment shows the potential of the proposed approach over the existing herb recommendation approaches.

**Weaknesses:**

* The long term effect of the items being recommended on the user has been studied under quite a few subarea in the recommender system.  Reinforcement learning (e.g., Interactive collaborative filtering), Counterfactual Reasoning (e.g., Unbiased learning-to-rank with biased feedback), as well as Evolving user profiles (e.g., Dynamic Poisson factorization) are some examples. Taking into long term effect is one of the main contribution claimed in the paper, yet the proposed module is not well positioned.
* Drug-Drug interaction is one of the major factor to be considered when performing drug recommendation (e.g, Wu et al., Conditional generation net for medication recommendation) in order to avoid harmful effects. The proposed approach does not take this information into consideration. As also show in the case study, the proposed proposed approach recommended quite a few herbs that does not matches doctors prescription, considering Drug-Drug interaction would be crucial for risk minimization.

**Questions:**

* How to position the proposed long term effect prediction module in related literature?
* How does the proposed approach mitigate the risk that would caused by harmful drug-drug interactions?

---

> ### Author Response · Authors · 2023-11-18
>
> **The reviewer's question regarding the long-term effects and the mention of various methods used in the recommender system field to consider long-term effects are greatly appreciated. We acknowledge the importance of these methods in modeling long-term effects in traditional recommender systems and their wide applicability across different domains and applications.**
>
>    However, it is essential to emphasize that TCM has unique characteristics in treatment methods and patient states. Therefore, specialized approaches are required to address long-term effects within this context. In our research, we are dedicated to addressing specific issues in the TCM recommendation domain. Consequently, our approach places particular emphasis on considering the long-term effects of herbs and the evolution of patient states.
>
>    We would like to clarify that our method is specifically designed for the TCM domain, and thus, it has different backgrounds and objectives when it comes to modeling long-term effects. In TCM treatment, the efficacy of herbal medicine often takes time to manifest and can be influenced by the overall condition of the patient as well as dynamic adjustments during treatment. Therefore, our method introduces the **Transition condition module** to dynamically capture changes in a patient's state between multiple diagnoses, allowing for more accurate herbal medicine recommendations. This module's design and application enable our model to consider the long-term effects on patients by dynamically integrating the patient's historical diagnosis information and current symptoms, resulting in more precise TCM recommendations. **Transition condition module, as stated in section 3.3 of the revised version, in line 203 of the paper, needs to be comprehensively understood in conjunction with the Horizontal Patient Representation.** We combine these two components to provide a detailed explanation of the impact of this module on long-term effects.** In Eq.(4),
> $\Phi^{(t)}_P = \boldsymbol{LSTM} \left( \left(  \Pi \left(h_S^{(t)},\Psi^{(t-1)}\right) ,C^{(t-1)} \right),...,\left( \Pi \left(h_S^{(0)},h_C^{(0)}\right) ,C^{(0)}\right); \; W_6\right)$
>
>  We employ LSTM for sequential prediction, where $\Phi^{(t)}$ denotes the comprehensive patient state during the $t$-th diagnosis. This state includes the patient's historical condition representation $h_C$, symptoms representation $h_S$, and the transitional state $\Phi^{(t-1)}$, derived dynamically through the state transition function
>
> $ \boldsymbol{T} \left(\Phi^{(t)}_P,\boldsymbol{P}\left(\Phi^{(t)}_P, E_h\right)\right) $
>
> $= \boldsymbol{NN_4} \left(Concatenate\left({\Phi_{P}^{(t)},(\Phi_{P}^{(t)}\odot  h_{H}^{(t)}),h_{H}^{(t)}}\right);\; W_8\right)$.
>
> **To offer a more intuitive illustration,** consider a scenario involving three diagnoses. During the first diagnosis, leveraging the patient's current condition representation $h_C^{(1)}$ and symptoms representation $h_S^{(1)}$, we compute the comprehensive patient state $Φ^{(1)}$, upon which the initial herbal recommendation is based. In subsequent diagnoses, the previous hidden state $C^{(1)}$ (retaining the first diagnosis's state information) and the transitional state $Ψ^{(1)}=T(Φ^{(1)}, hc^{(1)})$  are integrated through the transition function $T$, alongside the symptoms representation $h_S^{(2)}$ of the second diagnosis, to derive the second comprehensive patient state $Φ^{(2)}$. Importantly, the second comprehensive patient state $Φ^{(2)}$ contains information from the first diagnosis, facilitating dynamic herbal recommendations. This incorporation enriches the patient's inherent state information, amalgamating the current and preceding diagnosis states, along with the transitional state after herbs. Consequently, this augmentation enhances model performance, facilitating improved herbal recommendations.
>
>    Also, the concept of the patient state encapsulates a holistic representation of the patient's well-being, thereby affording a deeper understanding of their condition. **For instance,** a patient's description of "recent severe insomnia accompanied by mild headaches" may not be fully comprehended based solely on symptom representation. Our model predicts the transitional state to address the dynamic nature of herbal effects, considering patient-specific variations and herb interactions. This underpins our rationale for incorporating the transitional state even in cases where the patient's subsequent diagnosis patient state is unavailable, reaffirming our model's robustness in herbal recommendation.
>
> **In the ablation experiments section of Section 4.2 of the revised version, in line 309 of the paper.** We conducted ablation experiments pertaining to this module. As depicted in Fig.3, the performance of both variants lags behind that of SCEIKG, further validating the substantial contribution of this module.
>
> Thank you to the reviewers for their comments.

---

> ### Author Response · Authors · 2023-11-18
>
> **Reviewers have rightly pointed out the importance of considering drug-drug interactions (DDIs) in medication recommendation systems.** While SCEIKG does not currently model DDIs explicitly, it implicitly addresses the issue of herb compatibility through an iterative interaction knowledge graph, which includes relationships between different herbs. The graph helps avoid recommending mutually exclusive herbs.**In addition, we have reflected on the results of herb interactions by analyzing real doctors. We will provide you with further analyses.**
>
> In the motivation section of our paper, we highlight a significant and perplexing issue in TCM, which is the compatibility of herbal medicines, specifically, the interactions between different herbs when combined. However, the TCM treatment process encompasses a vast amount of complex TCM knowledge and involves four steps, making it challenging to elucidate the intricate interactions between different herbs. **To address this issue, we propose an approach based on domain knowledge from a knowledge graph to ensure herbal compatibility. Nevertheless, it's important to note that due to the complexity of TCM and the lack of standardization, accurately determining interactions between herbs can be challenging. For instance, herbs A and B may have antagonistic effects, but they might reach equilibrium when combined with herb C.** These interaction criteria are highly intricate, and herb compatibility is not fixed. Therefore, to enhance the accuracy of herbal recommendations, we constrain TCM recommendations based on domain knowledge from traditional Chinese medicine. Specifically, we introduce a regularization scheme
>  $\boldsymbol{R}$
> $=-\sum_{i=1}^{N}\sum_{j=1}^{N}W_{ij}^{h}\hat{hc}^{(t)}_i \hat{hc}^{(t)}_j$ to penalize $P^{(t)}$, limiting violations of herb pairs. The $W^{I}$ derived $W{i,j}^{h}$
> weights from $\mathcal{G}^{I}$ represent the strength of compatibility between the i-th and j-th herbs. If they are mutually antagonistic, $W{i,j}^{h}=0$.
>
> **We demonstrate that we reduce the risk of herbal incompatibility by both**
>
> First, in the Herb Compatibility section of Appendix E of this paper, on line 679 of the paper, we conduct a detailed result analysis. Due to data privacy and confidentiality, we present partial correlations between herb pairs in the appendix, which dynamically change during model training. According to real doctors' analysis, these changes are reasonable as they adapt to the patient's current condition.  **The reviewer suggests that the recommended results of our proposed method rarely agree with those recommended by real doctors, but we have analyzed them by real doctors and the remedies we recommend are also appropriate for the current patient's condition. In Table 8**, we showcase the results of herbal recommendations. Although there are differences between the herbs we recommend during the first diagnosis and those prescribed by doctors, **real doctors' analysis indicates that the model-recommended prescription is better suited to the patient's current condition. During the second diagnosis, although there are differences between the herbal recommendations and the actual prescription, doctors point out that the gap between them is not significant, and there are no mutually antagonistic herbs.** This further validates the effectiveness of considering herb compatibility and emphasizes that herb compatibility is a complex and dynamic issue.
>
> **Secondly, the experimental analyses were added to 4.2 in our revised version, in line 263 of the paper, highlighted in red. Fig. 4 illustrates the heatmap for all herb pairs; herb names are omitted due to data privacy. From the heatmap, we observe that the magnified positions, specifically (Cassia Bark(肉桂)), Red Halloysite(赤石脂)), and (Danshen Root(丹参), Lightyellow Sophora Root(苦参)), both exhibit a correlation of 0. The correlation between "Cassia Bark(肉桂)" and "Red Halloysite(赤石脂)" is 0, indicating a certain degree of mutual antagonism between these two herbs. This aligns with the ancient literature's viewpoint that "Cassia Bark is effective in regulating cold energy, but it loses its efficacy when encountered with Red Halloysite(官桂善能调冷气，若逢石脂便相欺)." In other words, Cinnamon and Red Halloysite are mutually repellent. Additionally, "Danshen Root(丹参)" and "Lightyellow Sophora Root(苦参)" cannot be mixed in some situations due to their differing medicinal properties.** Nevertheless, real doctors also point out that some herb combinations may be controversial as they may have different effects in clinical practice. We acknowledge that we have not yet fully addressed the issue of herb interactions, but we have selected **70** recommended results for real doctors to analyze, and after real doctors' analysis, we have achieved a **91.4%** level of compatibility for our recommended herb pairings, which, in terms of our recommended results, demonstrates the validity of our method.

---

> > ### Author Response · Authors · 2023-11-23
> > **Questions remaining?**
> >
> > Dear R 9xVT
> >
> > Does our response address your concerns? Do you have any remaining questions or concerns following our response? Please let us know. We’d be very happy to do anything we can that would be helpful in the time remaining! We hope the score could be raised if there is no further concern about our work. Thank you very much.
> >
> > The authors.

---

### Official Review · Reviewer_TUE4 · 2023-11-03

**Soundness:** 2 fair
**Presentation:** 2 fair
**Contribution:** 2 fair
**Rating:** 5
**Confidence:** 4

**Summary:**

This paper proposed a framework for traditional Chinese medicine recommendation. The proposed SCEIKG method leverages additional knowledge through incorporating IKG, and predicts the patients’ next visit state. As a result, it shows SOTA performance on a private dataset.

**Strengths:**

1.	The task of this paper to recommend traditional Chinese medicine with deep learning method is important.
2.	The idea of predicting next visit patient state is interesting and effective.
3.	The proposed method outperforms baselines on most metrics on a private dataset.

**Weaknesses:**

1.	The theoretical novelty is limited.
2.	The dataset for experiment is rather small.
3.	The presentation of method can be refined to make it easier to be understood.
4.	It’s not clear how the KG is completed and whether the predicted edges are reasonable. The effectiveness of incorporating IKG also needs to be clarified.

**Questions:**

1.	The font size in figures is generally too small. Especially Fig. 2, which is important for model understanding.
2.	It’s recommended to define loss function (L_{IKG}, L_{mse}, etc.) in 4.1-4.3 to make the method easier to be understood.
3.	In Fig. 3, method w/o IKG performs comparable with SCEIKG and even better on some metrics. Does the IKG really work?
4.	In Fig. 3, method w/o sequence performs very poor, which is confusing, as doctor can prescribe accurately even without any historical information. A successful AI system should also be robust to patient without historical information.
5.	If you can consider adverse drug-disease or drug-drug interaction, it will be a great contribution.
6.	It’s recommended to include metric ‘Jaccard’ to verify the set recommendation accuracy of the proposed method.

---

> ### Author Response · Authors · 2023-11-18
>
> 1. **The theoretical novelty is limited.**
>
> We sincerely appreciate your evaluation of our work. We are delighted that you have raised some questions, as this has prompted us to delve deeper into our research and make improvements. In response to your comments and concerns, we would like to further elaborate on the innovative aspects of our study.
>
>    We do acknowledge the limitations of the novelty of our research in terms of theory. **The novelty of our work lies in the introduction of the Interaction Knowledge Graph (IKG), which integrates domain knowledge from TCM through an interactive knowledge graph and applies it to sequential herbal medicine recommendations. This approach, combined with our transition condition module, which explicitly models the evolution of patients' conditions, sets our work apart from existing drug recommendation systems that may not capture these dynamic aspects​​.**
>
>    **Firstly: Incorporation of Domain Knowledge and Knowledge Graph Completion** We fully recognize that constructing a comprehensive knowledge graph for TCM is a great challenge that requires substantial data support. In our study, we employed knowledge graph completion to infer new facts from known facts, enriching the knowledge graph and providing more information for herbal medicine recommendations. Training the knowledge graph embeddings together with the network is distinct from traditional recommendation systems, which typically focus on modeling individual users and items. In our case, we need to consider symptom sets on one side and herb sets on the other, which vary in size and pose a unique challenge. This comprehensive approach allows us to more accurately represent and utilize domain-specific knowledge in the field of TCM.
> **Additionally, we conducted a supplementary Evaluation of IKG in Revision 4.2, at line 302 of the paper, where we experimented with various knowledge graph completion methods, and further confirmed the validity of our method by the results highlighted in orange in Table 3.**
>
>    **Secondly: Modeling Based on Sequential Data and State Transitions** We deeply understand the significance of modeling patient state transitions after herbs in the TCM. In our research, we introduced latent state representations to capture changes in patient conditions after taking prescriptions. This process is crucial in real-life scenarios, as doctors typically recommend herbs based on a patient's past diagnosis history and current condition. Our approach involves predicting patient states post-herbs and leveraging the potential inference in the state transition process. Even in cases where we lack comprehensive records of a patient's next diagnosis, our method allows us to predict a patient's conditional representation, enabling us to make reasonable herb recommendations. **This aspect, which has been less considered in prior research on sequence data-based medication recommendations, is pivotal for better recommendations**.
>
>    In summary, our research takes into account the fusion of domain knowledge and sequential data, enhancing the accuracy of herbal medicine recommendations. Once again, we are grateful for the reviewer's suggestions and questions, which have guided us toward deeper reflection and provided directions for further research.

---

> ### Author Response · Authors · 2023-11-18
>
> 2. **The dataset for the experiment is rather small.**
>
> Regarding the reviewer's concern about the small size of the experimental dataset, we would like to address the issue from the following three aspects.
>
> **Firstly,** we acknowledge that the dataset utilized in this paper is relatively small. Due to the subdivision of traditional Chinese medicine (TCM) and concerns related to patient privacy, the scale of TCM datasets is indeed constrained. However, the TCM sequence dataset we utilized is extremely valuable, accumulated over a decade from the Guangdong Provincial Hospital of Chinese Medicine, one of China's top hospitals. It encompasses patient information from various regions of China, not limited to Guangdong Province. Additionally, publicly available datasets for TCM recommendations, such as TCM, typically only include information on symptoms and herbs, making them relatively limited for research on TCM sequence recommendations. Therefore, despite the small dataset size, these data hold significant value in the context of TCM research and provide ample diversity for validating the SCEIKG model. Furthermore, we plan to gradually expand the dataset size while ensuring patient privacy and data quality as more data becomes available.
>
> **Secondly,** despite the small size of our dataset, during model training, we not only utilized the ZzzTCM dataset but also trained on the IKG. The knowledge graph we constructed contains 4,308,799 triplets, involving 344,092 entities and 35 types of edge relationships. Both the TCM dataset and the ZzzTCM dataset were incorporated into the construction of the IKG. **Detailed descriptions of the IKG are provided in Appendix A.1, and Appendix Table 4 outlines the data information,** indicating the relative richness of our data resources.
>
> **Finally,** we conducted additional experiments where we partitioned the ZzzTCM training data at different ratios to assess the robustness of our model in situations with a relatively small dataset. The results are presented in the table below,
>
> | TrainingDataset | Precision@5 | Precision@10 | Precision@20 | Recall@5 | Recall@10 | Recall@20 | F1@5 | F1@10 | F1@20 |
> |------------------|-------------|--------------|--------------|----------|-----------|-----------|------|-------|-------|
> | 0.2Train         | 0.4813      | 0.3919       | 0.2809       | 0.2454   | 0.3913    | 0.5514    | 0.3164 | 0.3803 | 0.3627 |
> | 0.4Train         | 0.4648      | 0.3847       | 0.2912       | 0.2352   | 0.3864    | 0.5722    | 0.3041 | 0.3741 | 0.3758 |
> | 0.6Train         | 0.4933      | 0.3960       | 0.2951       | 0.2571   | 0.4088    | 0.5933    | 0.3287 | 0.3901 | 0.3835 |
> | 0.8Train         | 0.4959      | 0.4025       | 0.2879       | 0.2551   | 0.4089    | 0.5733    | 0.3264 | 0.3931 | 0.3721 |
> | 0.9Train         | 0.5084      | 0.4206       | 0.3040       | 0.2470   | 0.4021    | 0.5718    | 0.3235 | 0.3994 | 0.3867 |
> | 0.94Train        | 0.5477      | 0.4275       | 0.3087       | 0.2727   | 0.4243    | 0.6010    | 0.3538 | 0.4128 | 0.3973 |
> | 0.98Train        | 0.6000      | 0.4579       | 0.3066       | 0.3059   | 0.4613    | 0.6151    | 0.3932 | 0.4465 | 0.3987 |
>
> From these results, it is evident that with an increase in the training set, the evaluation metrics show an upward trend to some extent. This indicates that the dataset size is not the sole determining factor for the model's performance. **We added this experiment to the Appendix C Parameter Sensitivity section of the revised version, at line 607 of the paper, and drew the trend of the results, displayed in Figures 6, 7, and 8 of the paper, and highlighted in red. See the revised version for details.**
>
> Once again, we appreciate your review and valuable feedback. We remain committed to making continuous improvements and refining our research.

---

> ### Author Response · Authors · 2023-11-18
>
> 3. **The presentation of the method can be refined to make it easier to understand.**
>
> We appreciate this feedback and will revise the manuscript to improve clarity and ease of understanding. We will provide more detailed explanations and simplify the terminology to make the approach more accessible to a wider audience. We have updated this section to take into account **Q1 and Q2** as commented by the reviewer, and the font size of the images has been changed, and we have highlighted in red in the revised version. **Furthermore, to facilitate a better understanding of our method, we would like to provide a further summary of our approach.**
>
>    Our model update process is similar to the concept of the EM algorithm. The updating of Interaction Knowledge Graph (IKG) is iteratively used to enhance various modules in Fig.2, and, in turn, the training of the model in Fig.2 improves IKG (on the right side of Fig.2). These two components collaborate, driving the entire iterative update process.**In our Appendix B, Algorithm 1 outlines the algorithm's process in detail.** Based on this, we will further explain the training process of the model within a single iteration.
>
>    Constructing a complete knowledge graph in TCM is a highly complex task that requires substantial data support. Therefore, knowledge graph completion from existing facts is crucial to make the knowledge graph more comprehensive and enrich entity information. Thus, in the first stage of **Algorithm 1 (Phase I: Interaction Knowledge Graph Completion),** we perform entity representation learning using **Eq. (3)** in the paper, where $f(h, r, t) = C * ||Sin(W_re_h + e_r - W_re_t)||_1$. Subsequently, we update it backward using the loss function **$L_{IKG}$ in Eq. (6)** to complete and enrich entity information.
>
>    In traditional knowledge graph embedding methods, typically only single knowledge triplets can be effectively represented, failing to capture higher-order similarities among entities. However, these higher-order relationships are critical for understanding complex interactions between entities, especially in the field of herbal medicine. For instance, ginger may treat both insomnia and cold symptoms, a multi-faceted relationship that cannot be accurately represented by a single triplet. To better aggregate symptom information and improve the quality of herb recommendations, we introduce the second stage of the algorithm. Building upon the first stage, we proceed to the second stage of **Algorithm 1 (Phase II: Recommended herbs based on sequential visits for each patient).**
>
>    In the 3.1 Section, we elaborate on how we employ a Graph Neural Network (GNN) approach to apply the entity knowledge representations obtained in the first stage to the second-stage herb recommendation. **In this process, we utilize an adjacency matrix constructed using the normalized pointwise mutual information method defined in Eq. (1) while incorporating the entity knowledge representations learned in the first stage. We then aggregate high-order relationships and high-order entity similarity information using Eq. (2):  $E^{(k)} = SUM(NN_1((E^{(k-1)}+W^{I}E^{(k-1)}W_1^{(k)}); W_3^{(k)}), NN_2((E^{(k-1)} ⊙ W^{I}E^{(k-1)}W_2^{(k)}); W_4^{(k)}))$.**  Finally, we propagate backward through the **loss function in Eq.(6)**, which includes the mean squared error **loss $L_{mse}$** to measure the distance between the recommended herb set and the actual herb set, the state **loss $L_{state}$** to assess the similarity between herb recommendations before and after, **and the regularization $\boldsymbol{R}$ to constrain the relationships between herbs.** Through iterative gradient descent, we optimize the model parameters to minimize the overall loss function, achieving the optimization goal of our model.
>
>    In summary, throughout the entire iterative process of the algorithm, we first learn entity representations for IKG, and then apply these representations to the second-stage herb recommendation model, as seen in the middle module of Fig.2. These two stages iteratively update each other, ultimately completing the training of the entire model. This process can be viewed as dynamic knowledge graph completion and herbal recommendation optimization. We thank the reviewers for their questions, which prompted us to delve deeper into the internal mechanisms and implementation details of the model.

---

> ### Author Response · Authors · 2023-11-18
>
> 4.  **Completion of KG**
>
> We describe the KG **in Appendix A.1 at the line 444 of the paper, Interaction Knowledge Graph**. Let us once again introduce the process of building the knowledge graph (KG). It primarily consists of two main stages. Firstly, we employ web scraping techniques using Python and other tools to dynamically fetch data from websites relevant to the domain. The relevant URLs are provided in Appendix Table 4. Subsequently, we systematically clean and organize this data. The second step involves applying manually designed rules to extract relevant triple knowledge from the web data. **For instance, we can transform information like "Chrysanthemum has the effect of promoting qi circulation and relieving pain, and can treat wind-heat cold" into representations of Traditional Chinese Medicine (TCM) knowledge, such as (Chrysanthemum, Medicinal Properties, Wind-Heat Cold) and (Chrysanthemum, Medicinal Functions, Promoting Qi Circulation and Relieving Pain). Through these steps, we construct the initial knowledge graph (KG).**
>
>    However, the initial KG may not encompass all the symptom and herbal entities present in the dataset.**To ensure the quality and comprehensiveness of the KG, we take further measures. We add the symptom and herbal entities mentioned in the dataset to the initial knowledge graph and build corresponding triples based on the dataset's content. For instance, (Symptom, Symptom-related Symptom, Symptom), (Symptom, Symptom-related Herbal, Herbal), and (Herbal, Herbal-related Herbal, Herbal) triples are constructed. This way, we form the final enhanced Interaction knowledge graph (IKG), as shown on the right side of Fig.2.**

---

> ### Author Response · Authors · 2023-11-18
>
> 5. **Rationality of IKG**
>
> **Regarding the reviewer's question about the rationale behind IKG, we would like to address it in conjunction with reviewer Q3.** IKG is an integral component of SCEIKG, consolidating Traditional Chinese Medicine (TCM) knowledge graphs and symptom-herb relationships. It is constructed using data from multiple sources and refined using the knowledge graph completion method. **We have analyzed this phenomenon in the Herb Recommendations section of Appendix E, in line 667 of the paper and highlighted in red.** And we will provide further analysis for your understanding. In addition, in order to facilitate a more intuitive observation of our results, we provide the following table for observations,
>
> | Sequential diagnoses | Symptom Set | Herb Set (w/o IKG) | Herb Set (SCEIKG) | Herb Set (TCM doctor) |
> |-----------------------|-------------|--------------------|-------------------|-----------------------|
> | First diagnosis       | 抑郁症(depression) | 黄芩(scutellaria baicalensis) | 黄芩(scutellaria baicalensis) | 黄芩(scutellaria baicalensis) |
> |                       | 口干(xerostomia) | 炙甘草(glycyrrhiza uralensis) | 炙甘草(glycyrrhiza uralensis) | 炙甘草(glycyrrhiza uralensis) |
> |                       | 大便费力(dyschezia) | 生姜(ginger) | 生姜(ginger) | 生姜(ginger) |
> |                       | 入睡困难(insomnia) | 大枣(jujube) | 大枣(jujube) | 大枣(jujube) |
> |                       | 眠浅易醒(light sleep, easy to wake up) | 人参(ginsen) | 人参(ginsen) | 人参(ginsen) |
> |                       | 乏力(fatigue) | 桂枝(cinnamomum cassia) | 桂枝(radix adenophorae) | 北沙参(radix adenophorae) |
> |                       | 胸闷(chest tightness) | 茯苓(tuckahoe) | 茯苓(bupleuri radix) | 柴胡(bupleuri radix) |
> |                       | 四肢麻木(numbness of limbs) | 川芎(sichuan lovage rhizome) | 白芍(paeonia lactiflora) | 天花粉(flos rosae rugosae) |
> |                       | 舌淡红(pale red tongue) | 法半夏(pinellia tuber) | 牡蛎(ostrea gigas) | |
> |                       | 下睑淡白(pale lower eyelid) | 当归(angelica sinensis (Oliv.) diels) | 干姜(zingiber officinale) | |
> | Second diagnosis      | 口干(xerostomia) | 黄芩(scutellaria baicalensis) | 黄芩(scutellaria baicalensis) | 黄芩(scutellaria baicalensis) |
> |                       | 惊恐(panic) | 赤芍(paeonia lactiflora) | 赤芍(paeonia lactiflora) | 赤芍(paeonia lactiflora) |
> |                       | 焦虑 (anxiety) | 炙甘草(glycyrrhiza uralensis) | 炙甘草(glycyrrhiza uralensis) | 炙甘草(glycyrrhiza uralensis) |
> |                       | 入睡困难(insomnia) | 大枣(jujube) | 大枣(jujube) | 大枣(jujube) |
> |                       | 眠浅易醒(light sleep, easy to wake up) | 生姜(ginger) | 生姜(ginger) | 生姜(ginger) |
> |                       | 乏力(fatigue) | 茯苓(tuckahoe) | 清半夏(pinellia tuber) | 清半夏(pinellia tuber) |
> |                       | 胸闷(chest tightness) | 人参(ginsen) | 人参(ginsen) | |
> |                       | 四肢麻木(numbness of limbs) | 桂枝(cinnamomum cassia) | 桂枝(cinnamomum cassia) | |
> |                       | 小便频急(frequent urination) | 甘草(glycyrrhiza uralensis fisch) | 茯苓(bupleuri radix) | |
> |                       | 右手心热(palm heat) | 当归(angelica sinensis (Oliv.) diels) | 炒六神曲(medicated leaven) | |
> |                       | 舌淡红(pale red tongue) | | | |
> |                       | 苔薄(thin fur) | | | |
> |                       | 下睑淡白边偏红 (pale lower eyelid with reddish edges) | | | |
>
> In above table, analyzed by real TCM doctor, **SCEIKG provides herbal recommendations more in line with classical prescriptions found in TCM ancient texts. In contrast, the model without IKG tends to favor prescriptions based on individual doctor's experiences. Typically, real doctors consider these classical prescriptions to be well-validated over the years and therefore more suitable in specific situations. This implies that our combinations may be more reasonable or in line with the historical usage of ancient formulas.** It also demonstrates that our IKG provides richer information, whereas models without IKG rely solely on the data itself for recommendations. This indirectly underscores the importance of IKG, offering more profound and comprehensive information for herbal medicine recommendations.
>
> **In addition, we added the recommended interpretability experiment in Appendix D of the revised version, at line 615 of the paper, highlighted it in orange, and gave Fig.11 for demonstration.** We use a patient and information fusion by the patient's symptoms to find multi-hop conenctivity based on the IKG as a way to justify the edges of our IKG. **See line 615 of the revised version for a detailed analysis.**

---

> ### Author Response · Authors · 2023-11-18
>
> 6.**performance w/o sequences**
>
> We highly appreciate the reviewer's perspective on the performance differences of our model described in Fig.3 in the absence of sequence information. We agree that a robust artificial intelligence system should demonstrate effectiveness in situations where extensive patient history data is not available, similar to what a doctor might do in practical medical practice. The performance in the absence of sequence data is indeed an area we aimed to explore to demonstrate the additional value of historical context in traditional Chinese medicine recommendations. Traditional Chinese medical practices often rely on a comprehensive understanding of a patient's condition over an extended period, including the evolution of symptoms and treatment responses. The primary innovation of our model lies in leveraging this sequential data to enhance accuracy.
>
> **However, the reviewer rightly points out that doctors can propose reasonable prescriptions even without such medical history. In response, we initiated further research. Through experiments, we discovered that when we did not utilize sequences for herbal recommendations, incorporating the overall patient condition into the model led to the results shown in Fig.3. Subsequently, we removed the overall patient condition and conducted experiments again using a model without sequence information, with the following results,**
>
>   | Ablation      | Precision@5 | Precision@10 | Precision@20 | Recall@5 | Recall@10 | Recall@20 | F1@5   | F1@10  | F1@20  |
> |-----------------------|-------------|--------------|---------------|----------|-----------|-----------|--------|--------|--------|
> |                       |             |              |               |          |           |           |        |        |        |
> | **w/o Sequence1**          | 0.2617      | 0.1490       | 0.1215        | 0.1249   | 0.1458    | 0.2252    | 0.1652 | 0.1432 | 0.1543 |
> | **w/o Sequence2**| 0.4282      | 0.3906       | 0.2872        | 0.2296   | 0.4053    | 0.5683    | 0.2872 | 0.3836 | 0.3708 |
>
>
> **w/o Sequence1** is the result of the ablation experiment of SCEIKG without sequence in our paper, and **w/o Sequence2** is likewise the result of the ablation experiment of SCEIKG without sequence, but with the overall condition of the patient removed from the experiment.
>
> The following analysis was conducted on the experimental results,
>
> **Role of the Sequence Module:** The sequence module may play a crucial role in handling sequential data, and removing it resulted in the model losing its ability to process sequence information. The information provided in the textual description may have been better integrated and utilized in the sequence module. In the absence of sequences, noise in the disease features may lead to interference from redundant information, thereby reducing performance.
>
> **Association between Features and Sequences:** The inclusion of overall patient condition as a feature in the model may depend on certain patterns or contextual information in the sequence data. Removing the sequence module may hinder the model's ability to correctly capture these associations, resulting in a decline in performance.
>
> **This part of the experiment we supplemented in Appendix D Poor Performance without sequences of the revised version, in line 645 of the paper and highlighted in red, see the revised version for more details.**
>
> We thank the reviewers for their constructive feedback, which prompted us to make valuable extensions to the study to better serve all potential clinical applications.

---

> ### Author Response · Authors · 2023-11-18
>
> 7.**drug-drug interaction**
>
> Thank you for raising these questions, we have considered them during the model's iteration to enhance its applicability and safety. While SCEIKG does not currently model DDIs explicitly, it implicitly addresses the issue of herb compatibility through an iterative interaction knowledge graph, which includes relationships between different herbs. The graph helps avoid recommending mutually exclusive herbs. **We have added a new experimental analysis of Herb Compatibility to section 4.2 of the revised version, which is demonstrated by Figure 4, in line 263 of the paper, and highlighted in red. In addition, we have reflected on the results of herb interactions by analyzing real doctors. We will provide you with further analyses.**
>
> In the motivation section of our paper, we highlight a significant and perplexing issue in TCM, which is the compatibility of herbal medicines, specifically, the interactions between different herbs when combined. However, the TCM treatment process encompasses a vast amount of complex TCM knowledge and involves four steps, making it challenging to elucidate the intricate interactions between different herbs. **To address this issue, we propose an approach based on domain knowledge from a knowledge graph to ensure herbal compatibility. Nevertheless, it's important to note that due to the complexity of TCM and the lack of standardization, accurately determining interactions between herbs can be challenging. For instance, herbs A and B may have antagonistic effects, but they might reach equilibrium when combined with herb C.** These interaction criteria are highly intricate, and herb compatibility is not fixed. Therefore, to enhance the accuracy of herbal recommendations, we constrain TCM recommendations based on domain knowledge from traditional Chinese medicine. Specifically, we introduce a regularization scheme
>     $\boldsymbol{R}$
>     $=-\sum_{i=1}^{N}\sum_{j=1}^{N}W_{ij}^{h}\hat{hc}^{(t)}_i \hat{hc}^{(t)}_j$ to penalize $P^{(t)}$, limiting violations of herb pairs. The $W^{I}$ derived $W{i,j}^{h}$
>     weights from $\mathcal{G}^{I}$ represent the strength of compatibility between the i-th and j-th herbs. If they are mutually exclusive, $W{i,j}^{h}=0.$
>
> **We demonstrate that we reduce the risk of herbal incompatibility by both**
>
> First, in the Herb Compatibility section of Appendix E of this paper, on line 679 of the paper. We performed the analysis of the results, and for data privacy and confidentiality reasons, we present in Appendix E some of the correlations between herb pairs that change dynamically during model training. Based on the analyses of actual physicians, these changes are justified as they adapt to the patient's current condition. Meanwhile,** in Table 8**, we showcase the results of herbal recommendations. Although there are differences between the herbs we recommend during the first diagnosis and those prescribed by doctors, **real doctors' analysis indicates that the model-recommended prescription is better suited to the patient's current condition. During the second diagnosis, although there are differences between the herbal recommendations and the actual prescription, doctors point out that the gap between them is not significant, and there are no mutually antagonistic herbs.** This further validates the effectiveness of considering herb compatibility and emphasizes that herb compatibility is a complex and dynamic issue.
>
> Secondly, **the experimental analyses were added to 4.2 in our revised version, in line 263 of the paper, highlighted in red.** Fig. 4 illustrates the heatmap for all herb pairs; herb names are omitted due to data privacy. From the heatmap, we observe that the magnified positions, specifically (Cassia Bark(肉桂)), Red Halloysite(赤石脂)), and (Danshen Root(丹参), Lightyellow Sophora Root(苦参)), both exhibit a correlation of 0. The correlation between "Cassia Bark(肉桂)" and "Red Halloysite(赤石脂)" is 0, indicating a certain degree of mutual antagonism between these two herbs. This aligns with the ancient literature's viewpoint that "Cassia Bark is effective in regulating cold energy, but it loses its efficacy when encountered with Red Halloysite(官桂善能调冷气，若逢石脂便相欺)." In other words, Cinnamon and Red Halloysite are mutually repellent. Additionally, "Danshen Root(丹参)" and "Lightyellow Sophora Root(苦参)" cannot be mixed in some situations due to their differing medicinal properties. Nevertheless, real doctors also point out that some herb combinations may be controversial as they may have different effects in clinical practice. **We acknowledge that we have not yet fully addressed the issue of herb interactions**, but we have selected **70** recommended results for real doctors to analyze, and after real doctors' analysis, we have achieved a **91.4%** level of compatibility for our recommended herb pairings, which, in terms of our recommended results, demonstrates the validity of our method.

---

> ### Author Response · Authors · 2023-11-18
>
> **8.metric Jaccard**
>
> Thanks to the reviewer's suggestion to include the Jaccard metric to verify the accuracy of set recommendation, we have conducted the experiments and obtained the results. The results are as follows，
>
> | Methods   | Precision@5 | Precision@10 | Precision@20 | Recall@5 | Recall@10 | Recall@20 | F1@5 | F1@10 | F1@20 | Jaccard |
> |-----------|-------------|--------------|---------------|----------|-----------|-----------|------|-------|-------|---------|
> | GAMENet   | 0.5066      | 0.4176       | 0.3096        | 0.2557   | 0.4151    | 0.6027    | 0.3300 | 0.4037 | 0.3976 | 0.1874  |
> | SafeDrug  | 0.5038      | 0.4082       | 0.3000          | 0.2562   | 0.4105    | 0.5926    | 0.2672 | 0.3364 | 0.3534 | 0.1791  |
> | Our       | 0.5477      | 0.4275       | 0.3087        | 0.2727   | 0.4243    | 0.6010     | 0.3538 | 0.4128 | 0.3973 | 0.2477  |
>
> We introduced the Jaccard metric to evaluate the set recommendation accuracy of our proposed method. Specifically, we compare the Jaccard similarity scores of the recommendations of our method with those of two methods, SafeDrug and GAMENet, which are also sequence recommendation methods, and the results are shown below, which also reflect the effectiveness of our model. We have added the experiment to the revised version, at line 256 of the paper, and highlighted it in red. We thank the reviewers for allowing us to further validate our model.

---

> > ### Author Response · Authors · 2023-11-23
> > **Questions remaining?**
> >
> > Dear R TEU4
> >
> > Does our response address your concerns? Do you have any remaining questions or concerns following our response? Please let us know. We’d be very happy to do anything we can that would be helpful in the time remaining! We hope the score could be raised if there is no further concern about our work. Thank you very much.
> >
> > The authors.

---

### Meta-Review · Area_Chair_s7Ds · 2023-12-10

**Metareview:**

This submission presents the Sequential Condition Evolved Interaction Knowledge Graph (SCEIKG) framework for recommending Traditional Chinese Medicine (TCM). The SCEIKG method integrates an interaction knowledge graph (IKG) to predict patients' next visit state, demonstrating state-of-the-art performance on a private dataset.

The reviewers have provided mixed feedbacks, with overall scores ranging from marginally below to marginally above the acceptance threshold. The key strengths and weaknesses identified by the reviewers are summarized below.

Strengths:
The paper addresses an important and novel task in the domain of TCM recommendation using deep learning methods.
The idea of predicting the next visit state of patients is innovative and has shown effectiveness.
The SCEIKG method demonstrates superior performance over baselines in most metrics on the utilized dataset.

Weaknesses:
The novelty of the approach is limited, and the method's presentation could be improved for clarity.
The dataset used for experiments is relatively small, raising concerns about the generalizability of the results.
There is a lack of clarity on how the knowledge graph is constructed and the rationale behind the predicted edges. The effectiveness of incorporating IKG is not thoroughly justified.
The paper does not adequately address drug-drug interactions, which is a critical aspect of TCM recommendation.
There are several presentation and formatting issues, including small font sizes in figures, unclear definitions, and inconsistencies in tables and figures.

The authors have made some adjustments to the presentation of the paper, but key concerns remain unaddressed. Reviewer TUE4 noted that despite improvements, the presentation issues, particularly in Figure 2, persist. Additionally, there are still errors and formatting issues that need rectification.

Given the mixed reviews and the authors' partial response to the concerns raised, the decision for this paper leans towards a weak reject. However, the importance of the task and the potential of the proposed method suggest that with significant improvements, particularly in addressing the novelty, dataset limitations, clarity in method presentation, and addressing drug-drug interactions, the paper could be a valuable contribution to the field. The authors are encouraged to thoroughly address these issues in future submissions.

**Justification For Why Not Higher Score:**

Numerous Weaknesses:
The novelty of the approach is limited, and the method's presentation could be improved for clarity.
The dataset used for experiments is relatively small, raising concerns about the generalizability of the results.
There is a lack of clarity on how the knowledge graph is constructed and the rationale behind the predicted edges. The effectiveness of incorporating IKG is not thoroughly justified.
The paper does not adequately address drug-drug interactions, which is a critical aspect of TCM recommendation.
There are several presentation and formatting issues, including small font sizes in figures, unclear definitions, and inconsistencies in tables and figures.

**Justification For Why Not Lower Score:**

N/A

---

### Decision · Program_Chairs · 2024-01-16

Reject